# Extant life detection using label-free video microscopy in analog aquatic environments

**Carl D. Snyder**[1☯], **Manuel Bedrossian**[2☯], **Casey Barr**[3‡], **Jody W. Deming**[4‡], **Chris A. Lindensmith**[5], **Christian Stenner**[6], **Jay L. Nadeau**[1☯*]

**1** Department of Physics, Portland State University, Portland, Oregon, United States of America, **2** Department of Medical Engineering, California Institute of Technology, Pasadena, California, United States of America, **3** Department of Earth Sciences, University of Southern California, Los Angeles, California, United States of America, **4** School of Oceanography, University of Washington, Seattle, Washington, United States of America, **5** Jet Propulsion Laboratory, California Institute of Technology, Pasadena, California, United States of America, **6** Alberta Speleological Society, Calgary, Canada

☯ These authors contributed equally to this work.
‡ These authors also contributed equally to this work.
* nadeau@pdx.edu

## Abstract

The ability of microbial active motion, morphology, and optical properties to serve as biosignatures was investigated by *in situ* video microscopy in a wide range of extreme field sites where such imaging had not been performed previously. These sites allowed for sampling seawater, sea ice brines, cryopeg brines, hypersaline pools and seeps, hyper-alkaline springs, and glaciovolcanic cave ice. In all samples except the cryopeg brine, active motion was observed without any sample treatment. Active motion was observed in the cryopeg brines when samples were subjected to a temperature gradient above *in situ*. In general, levels of motility were low in the field samples collected at temperatures < 4ºC. Non-motile cells could be distinguished from microminerals by differences in passive motion (e.g., density measured by sinking/floating), refractive index and/or absorbance, or morphology in the case of larger eukaryotes. Dramatic increases in the fraction of motile cells were seen with simple stimuli such as warming or the addition of L-serine. Chemotaxis and thermotaxis were also observed in select samples. An open-source, autonomous software package with computational requirements that can be scaled to spaceflight computers was used to classify the data. These results demonstrate the utility of volumetric light microscopy for life detection, but also suggest the importance of developing methods to stimulate cells *in situ* and process data using the restrictions imposed by mission bandwidth, as well as instruments to capture cell-like objects for detailed chemical analysis.

## 1. Introduction

Life is a collective phenomenon without a broadly accepted reductionist definition. Biosignatures can suggest the presence of past or present life through the observation of a phenomenon, characteristic, feature, element, molecule, substance, etc. A single biosignature (e.g., chemical) can suggest the presence of life, but in order to detect extant life conclusively, a relatively long list of life-defining features must be observed: metabolism, growth, reproduction,

**Data availability statement:** All raw hologram files and metadata for their reconstruction are available from the Dryad database (DOI: 10.5061/dryad.70rxwdc41) https://doi.org/10.5061/dryad.70rxwdc41https://doi.org/10.5061/dryad. 70rxwdc41.

**Funding:** Funding was provided by the National Science Foundation (1828793) and a National Aeronautics and Space Administration (NASA) PSTAR grant (80NSSC18K1738) for THOR (Thermal High-voltage Ocean-penetrator Research platform). Portions of this work were supported under a contract from, or performed at, the Jet Propulsion Laboratory, California Institute of Technology, under a contract with the National Aeronautics and Space Administration. Other portions were supported by grant ARC-1203267 from NSF and grant 5488 from the Gordon and Betty Moore Foundation (JWD). The funders had no role in study design, data collection and analysis, decision to publish, or preparation of the manuscript.

**Competing interests:** The authors declare no competing financial interests.

evolution, response to environment, and complexity (e.g., molecular, pathway, and morphological complexity) are usually included in the definition [1–3]. While all these features are to some degree necessary to define a lifeform, not all are shown by each individual (e.g., reproduction) or measurable on a timescale of a single lifetime (e.g., evolution). Techniques for identifying these features of life (i.e., biosignatures) are central to astrobiological goals of detecting and possibly studying life on other planetary bodies. This study investigates the ability of digital holographic microscopy to conduct *in situ* experiments to detect and classify the prevalence of active motion, morphology, and optical properties of microbial cells as biosignatures of extant life across a variety of extreme environments here on Earth.

Motility was the first accepted microbial biosignature. When Leeuwenhoek first observed bacteria and protozoa in the 17th century using single-lens microscopes, he knew they were alive because of their motion: "[N]o more pleasant sight has ever yet come before my eye than these many thousands of living creatures, seen all alive in a little drop of water, moving among one another, each several creature having its own proper motion" [4]. Meaningful, directed motion is a definitive biosignature that makes no assumptions about the chemical composition of the organisms under study [5]. As early as 1966, "Motion of a type that would not be expected for non-living systems" was suggested as a biosignature for Mars [6]. Active motion in this paper, as detailed in the methods section is characterized as any cell like object that is moving in a way that is distinct from any known abiotic motion (i.e., Brownian, fluid flow, buoyancy) [5,7].

With non-motile cells simple visualization of morphologically complex organisms in natural samples can often be sufficient to conclude that they are alive [5,8–10]. With smaller organisms —bacteria and archaea—the situation is more complicated. Their cells contain few features to suggest metabolism or complexity, but their optical properties, refractive index and absorbance, alter their contrast uniquely relative to most microminerals present in *in situ* samples [11]. Therefore, non-motile organisms can be classified by their distinct appearance due to their morphological features or optical properties relative to abiotic objects.

Microscopy of all kinds has been largely neglected as a tool for life detection in astrobiology, in part because there has been no NASA mission since Viking in 1976 that has attempted to find extant life beyond Earth. The Japanese Space Agency (JAXA) has developed a light microscope specifically for life detection in future missions, beginning with the JAXA mission *MMX*, which aims to return material from Phobos [12]. In 2014, when NASA initiated development of an extant-life detection mission to Europa, rapidly redefining appropriate instrument suites became necessary. The 2016 Europa Science Definition Team specifically called for a microscope capable of detecting organisms 0.2 μm or larger in diameter at densities of $10^3$ cells per mL or higher [3], without specifying instrument details. With the postponement of that mission, mission planning has turned to other sites, including Mars and Ocean Worlds such as Enceladus. Defining a suite of instruments that can detect microbial life is still very much a work in progress. While most Mars experts agree that the possibility of active microbial life in the Special Regions remains [13], existing flight instruments could not detect it at a level equivalent to what is found in Chilean desert analogs [14]. Studies have also emphasized the importance of multiple, complementary techniques for life detection, in which microscopy can play several roles in detecting cell structure and composition [15,16].

Using digital holographic microscopy (DHM), we imaged samples obtained from a cross-section of extreme environments, from hot deserts to the frozen Arctic and ultrabasic springs in coastal California, to determine the ability of the technique to identify extant life *in situ*. All samples were from aqueous or icy environments, representing possible analogs to Ocean Worlds. The purpose of collecting *in situ* data from a variety of extreme environments and stimulation experiments here on Earth is to start to build a framework for detecting

biosignatures with a DHM instrument for possible future extant life detection missions on a range of planetary body analog environments. All three microbial biosignatures we examined (i.e., active motion, morphology, and optical properties) are present in almost all sites and at least one is present in every site. The hypothesis of this work is that while not every microbe will exhibit detectable motion, morphology, or optical property biosignatures, within most environments – no matter how extreme – some fraction of microbes will be identifiable by these biosignatures. The fraction of motile cells may be increased by stimulation with heat [17], simple nutrients [18], salinity gradients [19], or other chemical or physical attractants or repellents [20,21].

## 1.1  Field sites and samples

Sites analyzed ranged across western North America and west Greenland (Fig 1). Field sites were visited that exhibited one or more environmental parameters that could be considered

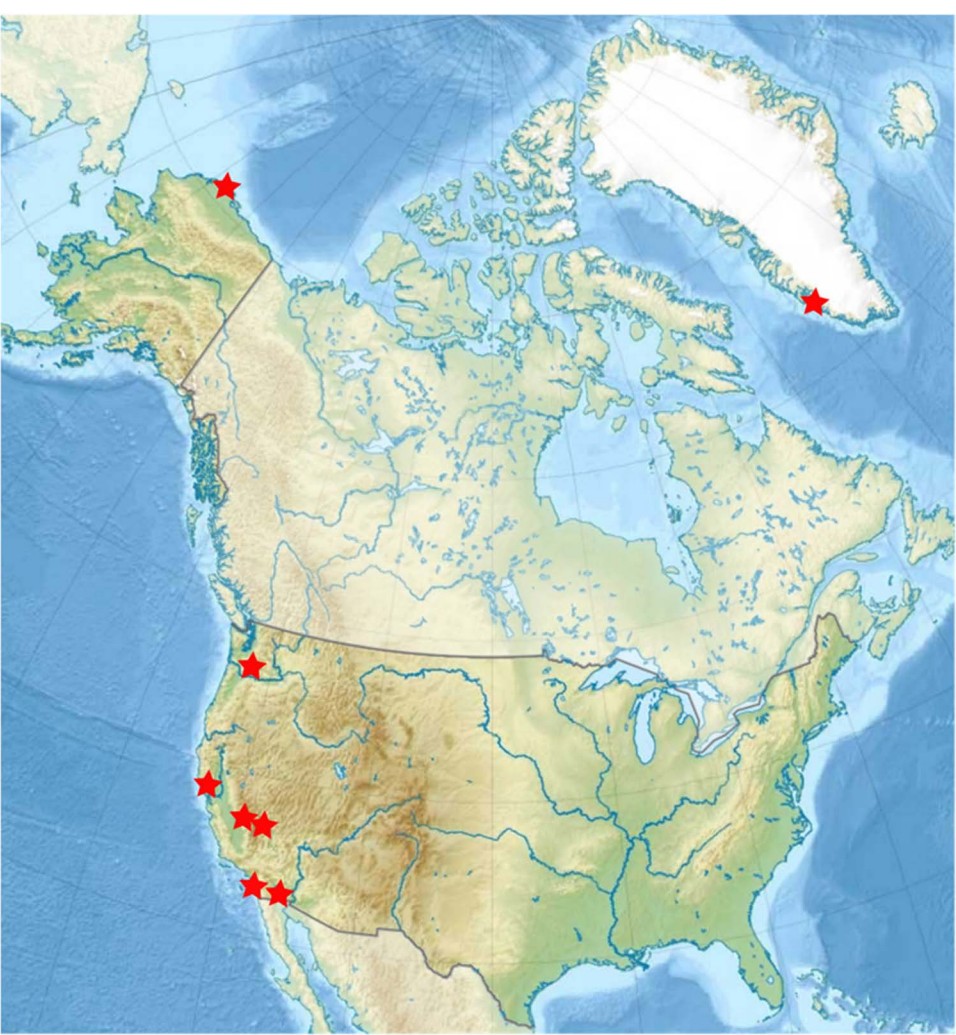

**Fig 1. Map of field site locations.** Locations include Newport Beach, CA; Malene Bay, Greenland; Utqiaġvik, AK; Salton Sea, CA; Badwater Basin Death Valley, CA; Ash Meadows, NV; Mt. St. Helens, WA; and The Cedars, CA. "North America laea relief location map" by Uwe Dedering is licensed under CC BY-SA 3.0. https://commons.wiki-media.org/wiki/File:North_America_laea_relief_location_map.jpg.

extreme: temperature, salinity, pH, oxygen concentration, or light availability. Most of them are known to be useful analogs for potential near-term lander missions, and others are emerging sites of interest. Tested were seawater, sea ice brines, cryopeg brines, glaciovolcanic cave ice, and water from thermal and hypersaline springs. Sea ice brine is liquid water that is trapped in micro-channels within the sea ice, with salinities depending upon temperature [22]. As seawater freezes, the crystalline structure of the ice that forms extrudes salt ions causing the nearby remaining liquid water to increase in solute concentration. This concentration increase depresses the freezing point of the remaining liquid, which keeps it in the liquid state [23]. Cryopegs are layers of marine sediment, saturated with brine, that became encased by permafrost many thousands (in some cases, millions) of years ago [24]. The brine has remained in liquid phase at subzero permafrost temperatures due to its high salinity (12–14%) [25]. Glaciovolcanic caves are a void space in ice or firn on ice mantled volcanoes, formed due to volcanic heat flux [26] Many of these sites or similar sites have been investigated to understand the microbes present, but to the best of our knowledge this paper presents the first results of microscopy or genetic analysis of microbes in ice from any volcanic subglacial void. A summary of the sites' extreme features and planetary body analogs are given in Table 1, and full descriptions with context images and references to geochemical analyses where available are provided in Supplementary Information text and S1–S5 Fig.

## 2.  Materials and methods

### 2.1  Digital holographic microscope (DHM)

The off-axis DHM used throughout this work has been described previously [48]. Its design has been optimized for easy adaptability to spaceflight (e.g., no compound objective lenses) and for balance between resolution and depth of field in order to obtain a limit of detection of ~ 100 bacteria per milliliter [49]. Detection of bacteria at this concentration meets the cells per milliliter detection requirements set by NASA's Europa Lander Study 2016 Report, making it

**Table 1.  Extreme environmental characteristics of the sites sampled.**

| Sample type | Site [Reference] | Extreme feature(s) | Analog [Reference] |
|---|---|---|---|
| Seawater | Newport Beach, CA | Temperate Earth seawater | None described |
| Seawater | Malene Bay, Greenland [27] | Cold temperature | Ocean Worlds |
| Seawater | Utqiaġvik, AK | Cold temperature | Ocean Worlds |
| Sea ice brine | Malene Bay, Greenland [28] | Subzero temperature, high salinity | Mars, Ocean Worlds [29] |
| Sea ice brine | Utqiaġvik, AK [30] | Subzero temperatures, high salinity | Mars, Ocean Worlds [29] |
| Cryopeg brine | Utqiaġvik, AK [24] | Subzero temperature, high salinity, millennia of isolation | Mars [31], Europa [32], Enceladus [33] |
| Hot saline spring | Salton Sea, CA [34,35] | High temperature, high salinity, low oxygen | None described |
| Hypersaline pool | Badwater Basin, CA [36] | High temperature, high salinity | Mars [31] |
| Freshwater spring | Ash Meadows, NV [37] | High temperature | None described |
| Glacier discharge | Mt. St. Helens, WA [38] | Low temperature, originating from ice | Mars, Io [39] |
| Hot spring/glacier mix, midstream, benthic mat | Mt. St. Helens, WA [38] | Interface of glacier ice and hydrothermal vent water | Mars, Io [39] |
| Hot spring/ glacier mix, downstream | Mt. St. Helens, WA [38] | Interface of glacier ice and hydrothermal vent water | Mars, Io [39] |
| Glaciovolcanic cave | Mt. St. Helens, WA [40] | Low temperature, lack of sunlight | Mars [41], Europa [42], Enceladus [43] |
| Barnes Spring 5, Hyperalkaline, low-sodium spring | The Cedars, CA [44] | High temperature, high pH, reduced oxygen | Mars [45,46], Enceladus [47] |
| Grotto Pool Spring 1, Hyperalkaline, low-sodium spring | The Cedars, CA [44] | High temperature, high salinity, high pH, low oxygen | Mars [45,46], Enceladus [47] |

a possible candidate for future life detection missions [3]. Briefly, a coherent and monochromatic light source (405 nm diode laser) is collimated. This light source is then passed through two microfluidic wells. One well contains a sample, while the other contains a reference liquid in order to match optical path lengths with the sample. The sample and reference beams are then passed through two separate identical aspheric objective lenses of numerical aperture (NA) = 0.3 and recombined at an image sensor via a relay lens. This instrument is capable of diffraction-limited sub-micrometer resolution. For field use, the optical train is housed in a rugged water-resistant container along with all necessary electronics for the instrument's stand-alone operation. These include a processor, hard drive, power source, laser, and camera, as well as diagnostic sensors. The acquisition speed is a maximum of 15 frames per second (fps). All videos linked in this paper are being played at 15 fps unless stated otherwise. The first iteration of the instrument, dubbed SHAMU, was first described in [27] and is pictured as deployed in Fig 2A, 2B. A smaller version was developed in 2019, referred to as Son of SHAMU, based upon a smaller pixel pitch camera (2.2 μm vs 3.45 μm), allowing correspondingly reduced relay optics that are folded to create a more compact system (pictures of deployments in Fig 2C, 2D). A comparison of the two instruments is shown in Supporting Information S6 Fig. The mass of the Son of SHAMU is 6 kg, reduced from 11 kg. Data are acquired using a custom open-source platform, DHMx [50]. Optical resolution of the two systems is the same.

## 2.2 Sampling and test conditions

Samples were collected from 8 different locations totaling 15 different sample sites across North America. Sites were chosen based on their wide range of aquatic environmental conditions allowing us to understand the ubiquity and detectability of the biosignatures of interest. Excluding Salton Sea, permission to sample was obtained from all sites. Salton Sea sample location was at a public beach access. Permission to collect samples in Greenland was obtained from the Greenland Climate Research Center. The issuing authority for research near

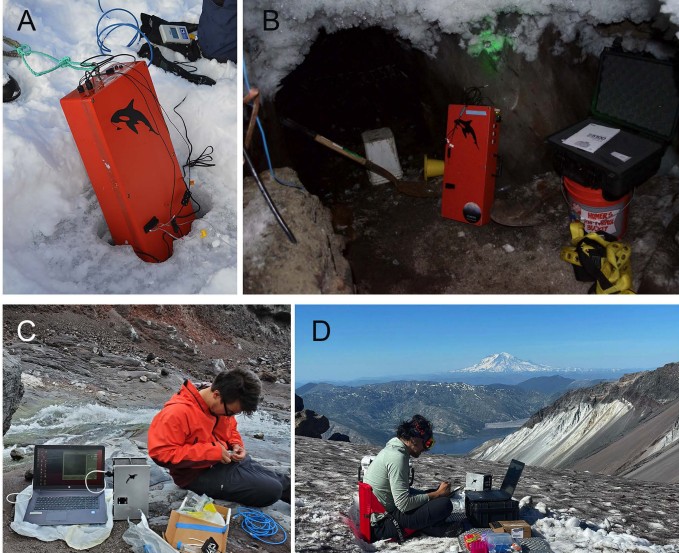

**Fig 2. Instrument deployments at field sites.** (A) Deployment in a sackhole of sea ice brine (Nuuk, Greenland). (B) Deployment in a permafrost tunnel providing borehole access to cryopegs (Utqiaġvik, AK). (C) Deployment in the Mt. St. Helens crater and (D) the Mt. St. Helens glacier caves with recordings performed at ambient air temperature.

Utqiaġvik, AK, was UIC Science, LLC. Field work in the Mt. St. Helens crater was performed under a U.S. Department of Agriculture Forest Service temporary special-use permit. The Cedar springs are located on private land held in a trust and permission was granted by Roger Raiche and David McCrory. The Newport beach work was approved by Caltech's Kerckhoff Marine Laboratory. Sample collection at Kings pool in Ash Meadows was under a special use permit granted by Ash Meadows National Wildlife Refuge. US National Park Service granted permission for sampling at the Badwater Basin Death Valley site. Locations for sampling generally were selected for their extreme conditions, making them viable candidates as astrobiological analog sites. To avoid contamination all samples were collected and stored in single-use sterile containers. For samples being transported, containers were stored in a secondary container that was passively cooled with ice when needed. All sample chambers used for viewing under microscopes were single-use and sterilized either by autoclaving or with ethanol wash during construction. These chambers were sealed from the exterior environment to avoid possible contamination. Transfer of samples from storage containers to sample chambers was done with single-use sterile syringes. More detailed descriptions of how different types of samples were collected can be found in the Supporting information.

## 2.3 Image processing

The aim of image processing was to improve the signal-to-noise ratio (SNR) of the particles to make them easier to identify, characterize as cell-like (or not), and track. A general image processing protocol can be seen in Fig 3. Until the last few decades all microscopy-based cellular classification and tracking was done manually by trained researchers [51]. Now that computers have become capable of single particle tracking and classifying motion, many more particles may be tracked at a time in some datasets, but manual tracking remains the most accurate method for discriminating particles from noise and linking tracks of a single particle [52–57]. No one approach works for all datasets, especially when sizes and speeds of particles vary and when samples represent a heterogeneous mix of particle types and noise. Thus, the process of reducing noise, localizing particles, and linking trajectories for our *in situ* experiments was unique for each recording, iterative, and a combination of manual and automated methods. We detail this carefully here in order to inform future similar work, as data processing techniques and details of manual tracking are often poorly described in the literature or tracking techniques are illustrated only for homogeneous populations with known particle shape and size.

Reconstruction of holograms into amplitude and phase images was performed with our previously published Fiji plug-ins [58] and the angular spectrum method, which has been shown to reduce noise relative to other methods such as Fresnel diffraction [59]. Spacing in z was chosen to reflect the size of the cells and the nominal axial resolution of the instrument (approximately 2-5 μm). Phase images were reconstructed using a reference hologram to remove aberrations [60]. Amplitude images were de-noised by a variety of methods depending upon the recording. Median subtraction at each z plane was always performed as a first step. If this approach was insufficient, band-pass filtering or application of a time derivative could be used to further reduce noise. After de-noising, a minimum or maximum z-projection was performed on the x, y, z, t hyperstack to yield an x, y, t stack in which all particles were projected onto the x, y plane.

The next step in the analysis pipeline was to process the x, y, t stacks with HELM, a software designed to characterize motility autonomously using realistic flight mission computational resources that has been characterized previously in detail [61–63]. The choice of amplitude or phase images for use in tracking also depended upon the recording. HELM

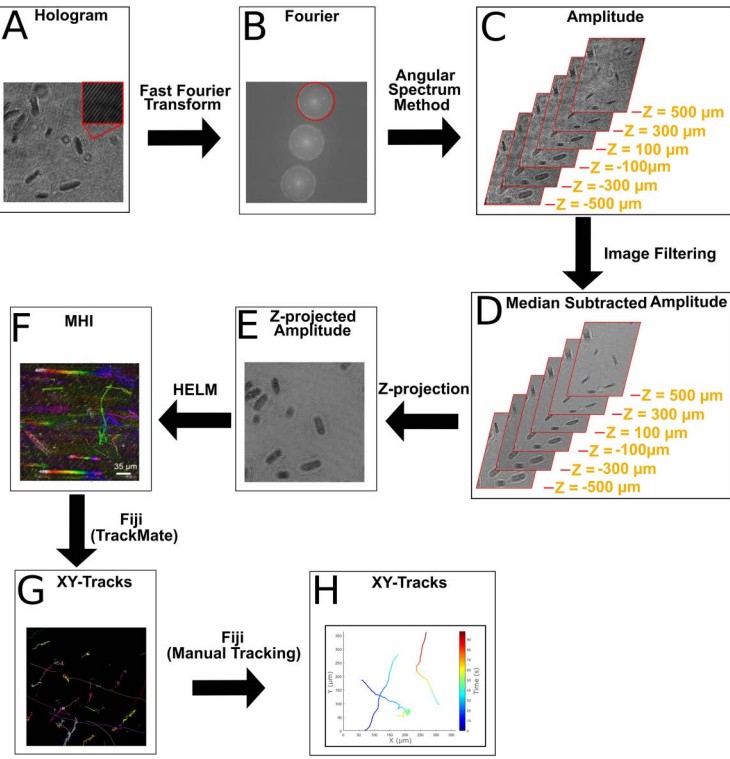

**Fig 3. Data processing flow diagram.** General data processing protocols used to characterize samples. A) Raw hologram with fringe pattern shown in magnified inset. B) The real or imaginary signal is selected in the Fourier plane, shown in the red circle. C) The angular spectrum method is used to reconstruct the data into multiple amplitude or phase z-planes. D) Several different image processing algorithms may be used on the reconstructed images; in most cases, median subtraction is sufficient. E) Z-projection may be used in sparse samples to obtain all of the 3D information in a 2D time series. F) A combination of automated and manual tracking is used to obtain tracks of motile cells.

identifies pixel changes in sequential DHM images, tracks clusters of change as particle movement, and classifies particles as motile or non-motile based on their movement patterns. These patterns are analyzed and classified by characterizing 23 track features (e.g., Brownian motion, fluid flow, etc), and then passing these features through a machine learning model [61]. A motion history image (MHI) summarizes a full video in one image by color mapping each pixel to the time index of largest intensity change; this image was used to quickly analyze whether motility was present in a sample and to help guide manual tracking. The x, y, t stacks used were usually z-projections over multiple planes, but occasionally better SNR was obtained using a single de-noised z-plane reconstruction rather than a projection. A researcher manually reviewed the HELM classifications to confirm accuracy and alongside this review process identified morphology and optical property biosignatures.

In datasets where false positive and false negative identifications by HELM's classifier were never below several dozen, moving particles were instead tracked using the Fiji plugin TrackMate, using LoG/DoG detection followed by Kalman or LAP tracking [53], or Manual Tracking. When drift was present, global drift was accounted for by tracking 3 non-motile particles (as indicated by the red arrows in S10 Fig) and subtracting their average instantaneous x, y velocities from the instantaneous corresponding x, y velocities of each motile organism (for more information regarding speed calculation and drift correction see Supporting information).

The cell concentration in the sample determined the number of frames that needed to be analyzed. One cell in a field of view corresponded to $\sim 10^4$ cells/mL [49], so in samples with concentrations of $10^5$ or greater motile cells/mL, a single 30s recording at 15 fps yielded sufficient motility statistics. Approximately 500 frames were tracked for such samples. In low biomass samples, more frames were needed to observe and track cells. For example, in Utqiaġvik seawater, over 3800 frames were analyzed: for cryopeg brine, over 16,000 frames.

## 2.4 Biosignatures

Trained experts manually identified and classified active motion, morphology, and optical properties as biosignatures through inspection of x, y, t stacks; x, y images; MHIs; and plotted tracks of particles. False positive rates for manual identification of biosignatures, while theoretically not zero, can be considered negligible [53,54].

**2.4.1 Active vs. passive motion.** Active motion is identified by distinguishing a particle's displacement from that caused by passive motion (e.g., Brownian, fluid flow, buoyancy) [7,61]. Fluid flow is readily characterized as unidirectional motion of all particles in the field of view, and may be subtracted readily by computer either separately or in conjunction with track analysis. HELM, as well as other particle trackers such as MOSAIC or NTA NanoTrackJ [64], distinguish Brownian motion by plotting root mean squared displacement $d$ vs. time for each tracked particle. Brownian motion scales with the square root of time t:

$$d = \sqrt{2Dt} \tag{1}$$

where D is the diffusion coefficient given as,

$$D = kT / 6\pi\eta r \tag{2}$$

In (2) $k$ is the Boltzmann constant, $T$ is temperature, $\eta$ is the viscosity of the fluid, and $r$ is the radius of the particle. Manual distinguishing of Brownian vs. active motion is based upon observation of displacement and is nearly always unambiguous as shown in Fig 4. At room temperature (25 ºC) with a 1 μm radius particle in water, the diffusion coefficient is $\sim 0.24$ μm²/s. At 4 ºC the viscosity of water is substantially greater, giving a diffusion coefficient of $\sim 0.13$ μm²/s. Measured bacterial diffusion coefficients can be 1000-2000 times greater than predicted when cells aggregate [65]. Making it important to ensure that aggregates be distinguished from single cells by imaging, tracking, or both. Rapidly swimming cells (v = 100 μm/s) displace much more rapidly than Brownian particles (Fig 4A). Even moderate swimming speeds of 10 μm/s are readily distinguished from Brownian motion of single cells; aggregates can be identified by displacement vs time curves or simply eliminated from manual tracks (Fig 4B). More slowly swimming cells are distinguished by their displacement vs. time curves (Fig 4C). We have previously studied Brownian motion of microparticles and non-motile cells in depth and found that displacements match the models well and that displacement vs time curves can be used to identify passive diffusion [7].

Along with directional drift and displacement-vs.-time curves, HELM also uses other track parameters such as sinuosity and angle of displacement to identify motile particles. In lab monoculture samples, the true positive rate over the false positive rate for automated motile track detection was reported to be 0.92 [61]. Manual results remain superior to any automated results, however, largely due to poor stitching of automatic tracks. Manual tracking combined with plotting displacement curves in the case of slow-moving cells essentially eliminates false positive identification of motile tracks.

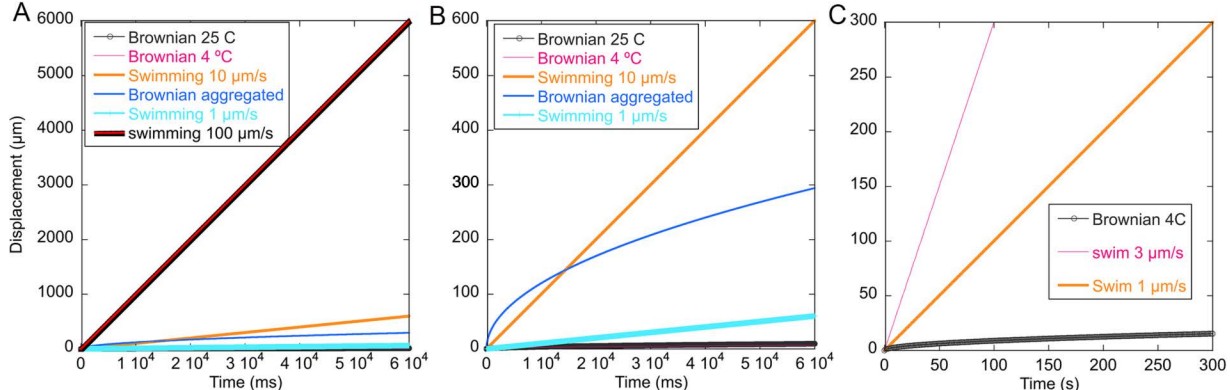

**Fig 4. Brownian motion vs swimming.** (A) Fast swimming, 100 μm/s, is clearly differentiated from Brownian motion even at short time scales of recording. (B) Moderate swimming speeds of 10 μm/s are still many times faster than Brownian motion of single cells and can be differentiated from Brownian motion of aggregates by displacement vs. time curves. (C) The slower swimming and diffusion rates seen at low temperatures can still be readily distinguished, although longer recordings are helpful to capture the shape of the curves.

The number of observed motile organisms in our samples was highly variable; some sites had only a few organisms identified over many recordings while some had dozens per field of view. This variability led us to use a qualitative analytical approach where motile organisms were classified as either highly abundant (multiple motile cells in each frame), abundant (at least one motile cell per frame), present (motile cells in some but not all frames), or not present (no motile cells in any frame).

**2.4.2 Morphology.** Morphological features can demonstrate a complexity that is an unambiguous biosignature. Morphology features include organelles, filamentous, appendages, and cellular clustering [5,9,10]. The resolution limit of our DHM instrument allows for the identification of organelles in larger eukaryotes. Cells identified to have organelles were classified as "large" cells and were generally > 20 μm in diameter. "Small" cells were those < 5 μm in diameter where no organelles could be distinguished, and cells could not be classified as eukaryotic or prokaryotic by DHM alone, as the environments studied often contained an abundance of small eukaryotes [66,67].

**2.4.3 Optical properties.** Absorbance and refractive index can both be used as a biosignature as both amplitude and phase images allow one to distinguish cells from microminerals. The difference in absorbance and refractive index of minerals, relative to cells, is generally represented in our microscopy images as higher contrast and sharper edges. Trained scientists can use these features alongside morphology and motion to label a particle as likely a cell or not.

Contrast in amplitude images is given by a combination of absorbance and scattering:

$$\sigma = \sigma_{abs} + \sigma_{scatt} \tag{3}$$

For absorbing particles, the absorbance term is approximately equal to the absorbance $A$ as calculated from the Beer-Lambert Law,

$$A = \varepsilon(\lambda)ct \tag{4}$$

where $\varepsilon(\lambda)$ is the extinction coefficient (or imaginary part of the refractive index, a measure of how much light is being absorbed at a given wavelength) of the absorbing molecule, $c$ is its concentration, and $t$ is the thickness of the cell. With our illumination wavelength at 405 nm,

chlorophyll is strongly absorbing, so that photosynthetic cells appeared dark. Many minerals, in contrast, are reflective rather than absorbing and appeared bright in amplitude. Small, nearly transparent cells such as bacteria are dominated by the scattering term, which can be approximated using Mie scattering theory or calculated more accurately for non-spherical cells using the discrete dipole approximation [68,69].

In phase reconstructions from DHM images, contrast $\Delta\phi$ is given by [70]

$$\Delta\phi = 2\pi\tau\Delta n / \lambda \tag{5}$$

where $\lambda$ is the wavelength of illumination, $t$ is the thickness of the specimen, and $\Delta n$ is the difference in (imaginary) refractive index between the object and its surrounding medium (in this case water, with $n = 1.33$). Most cells have $\Delta n \sim 0.05$, whereas minerals have values 5-10 fold higher [11]. The limit of detection with a single exposure on our system was approximately $\frac{\Delta\phi}{2\pi} = 0.05$, or $\frac{t}{\lambda} \sim 1$ for detection of cells.

## 2.5 Estimate of cell count by DHM

Cell counts were made by identifying one or more of each of the biosignatures described previously. Each volume of view in the DHM was 0.24 μL, so cell counts were estimated by averaging the number of cells per volume of view over the number of captured frames.

## 2.6 Motility stimulation via taxis

Microbial taxis is characterized by the orientation and movement of a cell along the gradient of a stimulant. *In situ* chemotaxis and thermotaxis experiments were performed for the purpose of understanding if adding a chemical or thermal gradient into a sample could increase the number of motile cells within the volume of view. Chemoattractants tested were L-serine, L-aspartate, and glucose and/or trehalose. Limited time and resources in the field led to chemotaxis experiments only being performed on samples from Newport Beach, Malene Bay, Utqiaġvik, Salton Sea, Death Valley, Ash Meadows, and thermotaxis experiments only being ran on Utqiaġvik, AK cryopeg samples. A higher concentration of motile cells within a recording after being exposed to chemical or thermal stimulants was characterized as a positive observation of chemotaxis or thermotaxis, respectively.

**2.6.1 Chemotaxis.** Chemotaxis was evaluated by creating a reservoir of the stimulant to be measured within a 1% agarose gel. Low-melting point (LMP) agarose (Sigma-Aldrich) was added to 0.9% saline and microwaved until dissolved. After cooling to ~60°C, chemoattractant was added to the desired concentration and the mixture was loaded into a sterile pulled borosilicate pipette tip or 27-gauge needle. This glass pipette was pulled on a Sutter Instruments P-97 puller using "patch clamp" settings. The tip diameter range was 1-5 μm. For experiments conducted in the field where small diameter glass pipettes were too fragile, sterile 27-gauge syringe needles were used to provide the chemoattractant substrate. After the agarose hardened, the substrate was inserted into the sample chamber for imaging.

**2.6.2 Thermotaxis.** The thermal gradient was produced using a proportional-integral-derivative (PID) temperature controller (Thorlabs TC200) with an embedded thermal sensor attached to the sample chamber via adhesive on one side of the controller. The heating element is attached to a thin metallic surface that is then thermally insulated in all directions except for the surface where the sample chamber sits. This creates a linear thermal gradient by constraining the flow of thermal energy to only one surface. While convection still occurs introducing non-linearity to the temperature gradient, these effects are negligible [71].

Samples were allowed to equilibrate for 3-5 min before confirming temperature gradients, and DHM recordings were collected at corresponding points in the sample chamber.

## 2.7 Ground truthing

Ground truthing included high resolution fluorescence imaging using dye staining and/or genetic sequencing. Both of these types of ground truthing data were collected for samples from seawater, sea ice brine, and cryopeg samples from Utqiaġvik, AK. Fluorescence microscopy only was performed on the other sea water sites, sea ice brine, hot spring, and glacial melt samples. Gene sequencing data only was used to compare cellular populations of the Cedar hyperalkaline spring systems and Death Valley Badwater basin pool with DHM data. No ground truthing data were collected for ice from subglacial void in Mount Saint Helens or Ash Meadows King pool. Having no ground truthing data to compare to DHM on a life detection mission is very possible and so we believe it is important to include results where we have no ground truthing data. Additionally, the Mountain Saint Helens data is the first study of microbial biosignatures present in ice core samples of any glaciovolcanic environment, and observation of potential life there should inspire "return missions" to the site.

**2.7.1 Live/dead staining fluorescence microscopy.** At some sites, samples were fixed immediately in 3% paraformaldehyde, either unstained or stained. Staining was performed with 0.1-0.5 μM SYTO9 or with a mixture of 0.2 μM SYTO9 and propidium iodide ("Live/Dead") (Fisher); staining was allowed to proceed for 5 min before fixation, and dyes were not removed. Widefield fluorescence and brightfield microscopy were performed using an Olympus IX71 inverted microscope with a 40x objective (numerical aperture 0.7). Fluorescence emission was collected using Hg lamp excitation with a multiband filter set (Chroma multiband filter #89402, 391-32/479-33/554-24/638-31) or sequentially with the enhanced green fluorescent protein/Texas Red filter sets and captured on a Zeiss Axiocam 305 RGB camera. Data processing was performed using Zeiss Zen. Cell counts were performed in a bacterial Petroff-Hausser cell counting chamber (Electron Microscopy Sciences) using both unstained samples (counted under brightfield illumination) or stained samples (counted under epifluorescence illumination). Cell counts were conducted by extracting 10 μL of live or fixed samples using a micropipette with a sterile pipette tip and transferring them into the chamber. Counts were averaged over at least 20 grid squares.

## 2.8 Chemical analysis

Standard chemical analyses were performed on all water samples. Full methods and data are available in the references for sites that have been reported previously.

## 3. Results

### 3.1 Sea water/sea ice brines - Nuuk, GL/Utqiaġvik, AK/Newport Beach, CA

In seawater and sea ice brines from Nuuk, Greenland, and Utqiaġvik, AK, as well as seawater from Newport Beach, CA, large microbial cells larger than 2 μm in the smallest dimension were seen in all samples. In all cases these cells were diverse, representing diatoms, photosynthetic flagellates, and ciliates (Fig 5A–C). Motile large and small microbes were observed in all *in situ* seawater and sea ice brine samples (S1–S4 Video), excluding the sea ice brine samples from Greenland, which only had spontaneously motile large microbial cells (S5 Video). The majority of spontaneously motile cells seen in all seawater and sea ice samples were small (2-10 μm) photosynthetic cells with mean speeds up to 50 μm/s, where chlorophyll-containing cells could be identified by their dark contrast at the 405 nm illumination wavelength. The most reliable method to stimulate small microbial motility, especially in samples collected at

subzero temperatures, was warming to 4°C. For Greenland samples in seawater, overnight incubation at 4°C with full culture medium (2216 marine broth) or with 1 mM L-serine resulted in large numbers of motile small microbes with speeds up to 100 μm/s. All seawater and sea ice samples examined showed small microbial run-reverse motility patterns indicating single polar flagella, consistent with the literature [72,73] (Fig 5D). Select motile organisms and their morphologies are shown in S7 Fig.

Counts of putative cells in sea ice brines from Greenland and Alaska showed about 100 cells per volume of view, or approximately 4 x $10^5$ cells/mL. Ground-truth measurements in the Greenland samples, determined by epifluorescence microscopy, showed a mean value of 3.03 [± 1.25] x $10^5$ cells $mL^{-1}$, N = 19, consistent with DHM results (Lindensmith *et al.*, 2016). Other seawater cell counts were comparable: Newport Beach, $10^3$ eukaryotes/mL and $10^5$ prokaryotes/mL [74]; Utqiaġvik, AK, $10^1$ eukaryotes/mL and $10^4$ prokaryotes/mL [75].

For samples in which nutrient addition resulted in increased motility, we analyzed chemotaxis towards a gradient or point source of nutrients, including L-serine, L-aspartate, and glucose and/or trehalose. All samples tested from seawater or sea ice showed chemotaxis towards L-serine, but not all samples collected from other environments. While chemotaxis has been demonstrated using microbes present in sea-ice brine conditions [27,76], no chemotaxis or increased motility was seen here with glucose or trehalose supplementation or gradients.

## 3.2 Hot spring – Salton Sea, CA

The sampled pool was at a temperature of 39°C, pH 6.9, and contained 3.4% NaCl, suggesting that the source water was partially derived from the Salton Sea. Notably, the ammonium concentration in the water was 28 mM and the sulfate concentration was 41 mM. This sample showed a large variety of small and large microbes; many of the large microbes showed elongated cell shapes often connected together into filaments, without the dark absorption characteristic of chlorophyll (Fig 6A–C).

No motility was apparent in the recordings at room temperature (22°C) (S6 Video). However, as the sample were warmed, first small microbial and then large microbial motility became apparent. At 25°C, a few small microbes were observed swimming. Approximately 1 motile cell per imaging volume or ~ 10% of micrometer-sized particles were motile (Fig 6D, S7 Video). At 30°C, the overall fraction of motile organisms increased markedly, to approximately 40% of particles and 4-5 per imaging volume (Fig 6E, S8 Video). At 42°C, a majority of

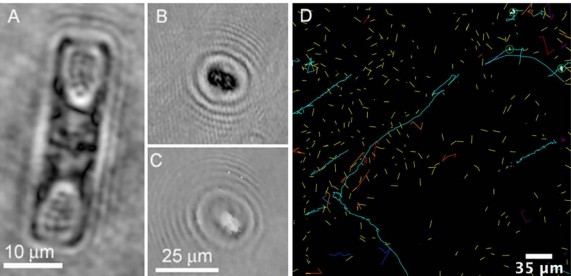

**Fig 5. Biosignatures from sea ice brine and seawater.** (A) Amplitude image of non-motile diatom from Greenland sea ice brine. Note the demarcation of cell walls and organelles. (B, C) Rapidly swimming organism in amplitude (B) and phase (C) from Greenland sea ice brine (images A-C previously published in [27]). (D) Motile tracks of small microbes from Greenland seawater incubated overnight at 4°C. Total duration = 300 frames or 20 s. Tracks were obtained from a sum of amplitude reconstructions taken every 5 μm over 500 μm depth and tracked using TrackMate automated detection and LAP tracking.

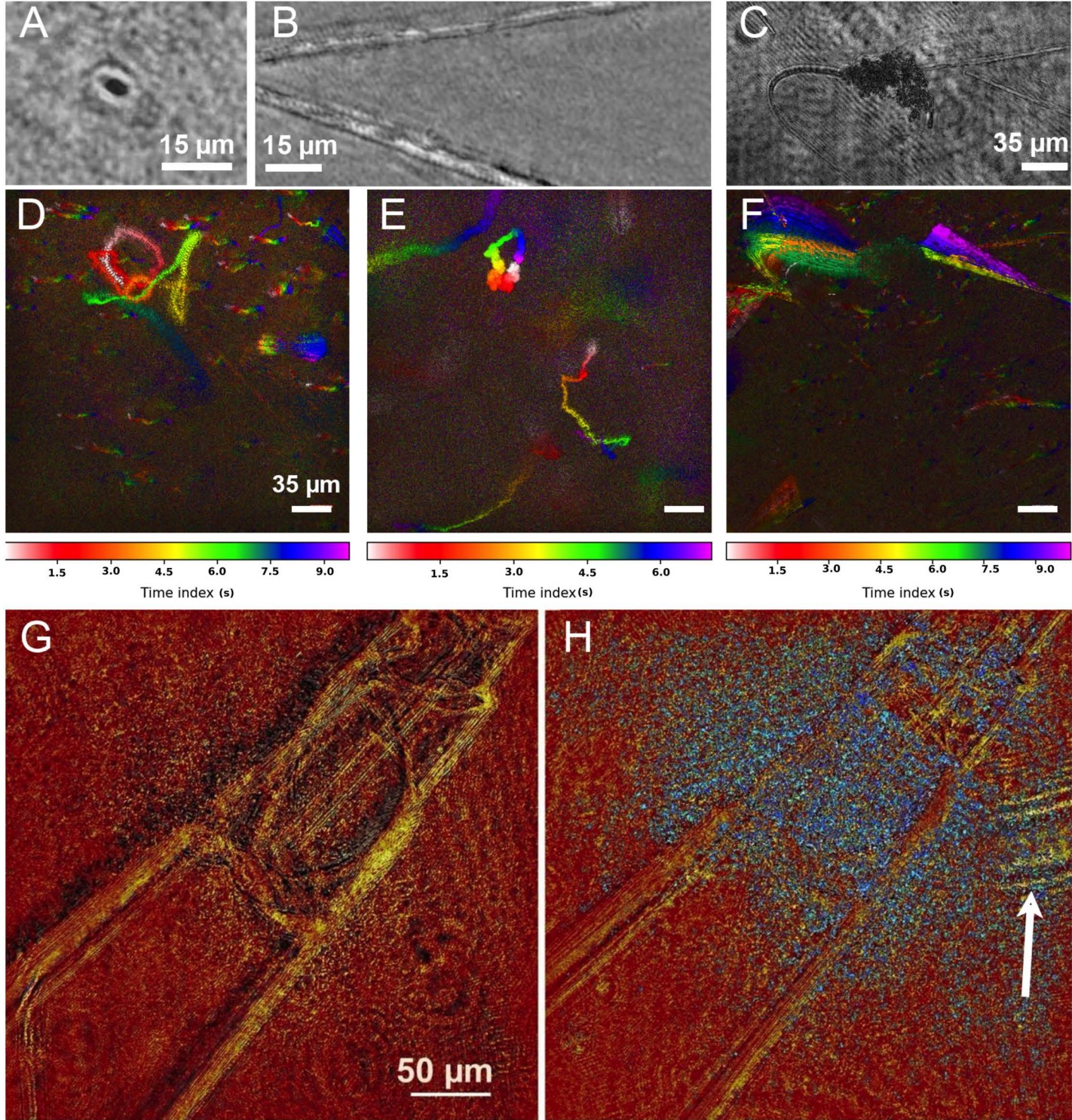

**Fig 6. Biosignatures in the Salton Sea hot spring.** (A-C) Amplitude reconstructions of large microbes. (A) Large microbe with dark absorption characteristic of chlorophyll. (B) Elongated filaments without chlorophyll. (C) Arrangement of several filaments with a cloud of putative cells/organics. (D-F) HELM motion history images (MHI) with color-coding indicating time lapse of selected areas at different temperatures. (D) At 25°C, a single motile small microbe is observed making a complex trajectory in the imaging volume over a 675 frame (45 s) recording period. (E) At least 3 highly motile small microbes are seen in 35 s at 30°C. (F) At 42°C, the large microbes are highly active, though moving slowly as this 45 s recording period shows (the upper part of this panel corresponds to Panel C). (G, H) Chemo-taxis and possible predation. (G) A needle filled with serine at the beginning of the recording. (H) Panel G after 60 min. Blue is a maximum projection of the frame-to-frame changes over the recording period. The arrow indicates a motile large microbe possibly preying on the small microbial cloud.

both small and large microbes showed motility. The non-photosynthetic large microbes began to demonstrate grazing behavior and to deform circularly and glide back and forth. The speed of the large microbes' motion was relatively slow, with about 40 μm of displacement (bending) observed over 45 s (Fig 6F, S9 Video).

Chemotaxis towards serine was observed in time lapse. A cloud of small microbes was visible around the needle tip within 5 min, with increased accumulation readily apparent after 60 min. Large microbes were observed likely preying on this microbial cloud (Fig 6G, 6H). Again, the motion of the small microbe was on the order of 1 μm/s and only readily apparent with 30-60 min of imaging (S10 Video). Chemotaxis towards glucose or $NH_4Cl$ was not observed with these organisms (not shown).

Ground-truthing using fluorescence microscopy confirmed that most elongated cells and filaments seen were not chlorophyll-containing. Minimal autofluorescence was seen from these cells or from small microbes; both became apparent under SYTO9 staining (S8A & S8B Fig). Filamentous algae were rarer and showed distinctly different morphologies, as well as intense red fluorescence indicating that they were photosynthetic (S8C Fig). Other single-celled photosynthetic organisms were also present (S8D Fig).

### 3.3 Desert spring system - Badwater Basin Death Valley, CA/Ash Meadows, NV

A full study of the Death Valley site, including metagenomic analysis, has been published [36]. Samples analyzed from Badwater Basin and Ash Meadows showed a predominance of small microbes, although a few large microbes were observed (Fig 7A, 7B). *In situ* recordings from the Badwater Basin pool showed 1-4 motile small microbes and 0-1 large microbes per imaging volume (S11 Video). Fig 7C, 7D shows data from an *in situ* recording of the tracking of two small microbial organisms demonstrating run-reverse swimming pattern (S12 Video). No chemotaxis towards L-serine was evident. Streptococci, filaments, spiral, and other morphological features were identified within the Badwater basin pool DHM data. As detailed in the full study of the Death Valley site microbes present in these samples, identified by 16sRNA sequencing, are known to have these same morphological features [36].

Recordings of Ash Meadows at roughly 20°C in the lab showed ~ 3 motile small microbes and 0-1 large microbes per imaging volume (S13 Video). The average speed [± SD] of the

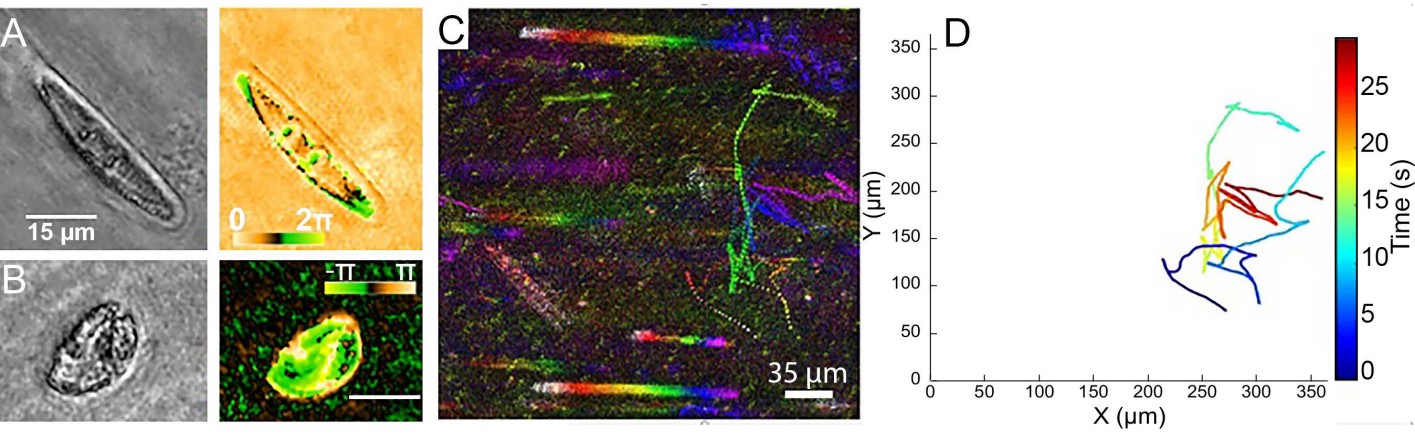

**Fig 7. Biosignatures from Death Valley/Ash Meadows.** (A) Amplitude (left) and phase (right) images of a motile (gliding) diatom from Badwater Basin Pool. (B) Amplitude (left) and phase (right) images of a motile ciliate from Ash Meadows. (C) MHI of an *in situ* recording showing the run-reverse swimming pattern of two organisms from Badwater basin in Death Valley. (D) Manually tracked trajectories of the motile organisms in (C).

motile cells was 66 [± 41] μm/s, N = 7. After heating to 30°C for 20 min the average speed of the recordings was 28 [± 19] μm/s, N = 8, and the number of motile cells per imaging volume was ~4 for both the room temperature and heated sample. No significant chemotaxis to L-serine was observed.

### 3.4  Glaciovolcanic: Glacial melt and hot spring – Mt. St. Helens Crater, WA

Samples were collected starting at the edge of the glacier and continuing downstream as shown in S4 Fig. Here we will refer to the sites as "Glacier water," "Midstream" and "Downstream." Full descriptions and geochemical analysis have been published [38]. Meltwater emerged from the glacier and was sampled ~100 m downstream at a temperature of 0.7°C. The glacier samples showed very few cells (approximately 1 per frame average, or ~$10^4$/mL) and contained a large amount of silt and microminerals (Fig 8). Larger mineral particles, mostly elongated or spindle-shaped, sank immediately to the bottom of the chamber (Fig 8A). Smaller particles remained suspended, and a significant fraction were motile (S14 Video). Distinguishing small, round particles from nonmotile microorganisms was sometimes ambiguous, although the motile particles had significantly less amplitude and phase contrast than most of the nonmotile ones, indicating that most of the nonmotile ones were likely microminerals (Fig 8B, 8C). When the particles were resolved (> 1 μm diameter), minerals could be identified by their irregular, faceted morphologies (Fig 8C); nearly all motile particles were ~1 μm in size. Based on epifluorescence microscopy the concentration of cells within glacier discharge samples was 3.4 [± 0.2] x $10^4$ cells mL$^{-1}$ [38].

At the midstream site, approximately halfway between the glacier surface discharge and downstream site, hot spring water mixed with glacier discharge to create pools at 30-37°C. These pools were decorated with orange-to-red mineral deposits and thick green microbial mats. Recordings were collected at ambient air temperature (~20°C). DHM recordings of the orange deposits showed motile small microbes that clearly differed in refractive index from the surrounding microminerals (Fig 9A–9C) (S15). While this feature was sufficient to distinguish minerals from cells, the density of the microminerals was also high, causing them to sink over a time course of tens of seconds (not shown). Tracking was challenging due to the background, but tracks could be obtained from frame-to-frame subtractions on reconstructions of single focal planes. No large microbes were seen in the orange regions. Under microscopy the green microbial mats showed

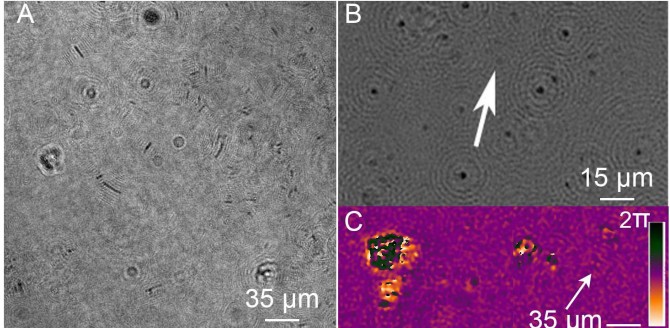

**Fig 8. Biosignatures from glacier meltwater from Crater Glacier, Mt. St. Helens.** (A) Spindle-shaped microminerals, amplitude reconstruction on the focal plane of the bottom of the sample chamber. (B) Amplitude reconstruction 80 μm above the bottom of the chamber, showing a motile organism (arrow) and some suspended non-motile particles with significantly higher contrast (S14 Video). (C) Phase image 80 μm above the bottom of the chamber, showing a motile organism (arrow) with low phase contrast surrounded by microminerals of high contrast.

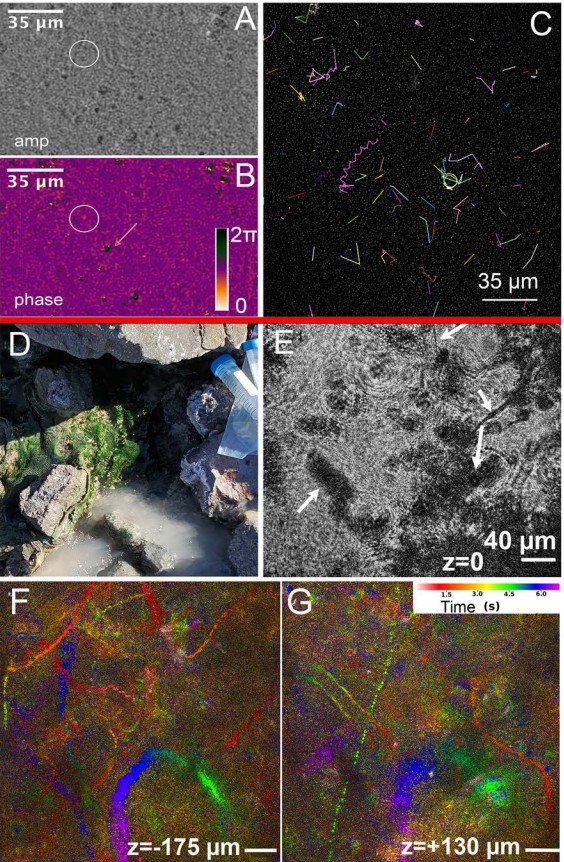

**Fig 9. Biosignatures from the Mt. St.** Helens crater. (A-C) Orange biofilm sludge. (A) Orange biofilm sludge amplitude reconstruction; the circle indicates a motile cell. (B) Orange biofilm sludge phase reconstruction, showing the phase shift contrast between the motile cell (circled) and microminerals (arrow). (C) 2D tracks of frame-to-frame subtracted reconstruction of glacier discharge showing several motile small microbes. (D-G) Midstream. (D) Context photo of photosynthetic cyanobacterial benthic mats. (E) Benthic mat DHM amplitude reconstruction at **z** = 0 (approximately halfway through the 1 mm deep sample chamber). (F, G) MHI at two different depths from the benthic mat, showing rapidly swimming organisms at very different focal planes.

an abundance of non-motile cyanobacteria as well as large numbers of rapidly swimming small microbes. The biofilms were apparent at the meter to centimeter scale (Fig 9D). Under DHM, the cyanobacteria could be seen, along with a high density of motile organisms swimming outside the plane of the mat (Fig 9E–9G) (S16 Video). Ground-truthing with fluorescence microscopy showed dense autofluorescent mats of filamentous cyanobacteria (S9 Fig). At the downstream site the recordings were dominated by single-celled large microbes. Both rapid swimming and Brownian motion with drift were observed; tracking allowed for analysis of both swimming speed and related parameters (not shown). Tracking on this dataset could be obtained after thresholding a single-plane reconstruction (S10 Fig). Epifluorescence microscopy showed cell concentrations of 5.6 [± 1.2] x $10^4$ cells mL$^{-1}$ [38].

## 3.5 Glaciovolcanic: Ice from subglacial void – Crater Glacier Mt. St. Helens, WA

*In situ* imaging of the glaciovolcanic cave ice samples showed low biomass, and only small microbial organisms were observed. Four of the seven recordings from the glacial ice samples

contained motile organisms, and in these recordings between 1 and 5 motile small microbes were detected per imaging volume (S17 Video). In the other three recordings no motility was detected. In Fig 10 we show data from one of the recordings where 6 motile small microbial organisms were detected. The mean speed [± SD] of the motile microbes from this recording was 8 [±2] μm/s, N = 6. The slowest of these motile cell moved with an average speed of over 4 μm/s, making its motility still distinguishable from Brownian motion. Motile organisms that were observed in the Mt. St. Helens glacial samples were extremely low signal-to-noise ratio (SNR), making identifying motile organisms in the MHI images extremely difficult. Fig 10A, 10B shows two motile organisms with the highest SNR of these datasets. Most of the motile organisms were not moving with any distinctive swimming patterns, but one appeared to be moving in a spiral or corkscrew pattern, as seen in the bottom left corner of Fig 10A, 10D and the zoomed-in trajectory in Fig 10C. The other motile organism that was seen in the MHI images is shown in Fig 10B, 10D; its trajectory consisted of a long run with no clear abrupt turns.

### 3.6 Hyperalkaline spring system - The Cedars, CA

Samples were collected directly from a series of hyperalkaline and low saline headsprings within The Cedars serpentinization spring system in coastal northern California; context images of these individual pools are shown in S3 Fig and are known as Barnes Spring 5 (BS5) and Grotto Pool Spring (GPS1). Even though both pools come from the same spring system, the pools vary in pH and redox potential. GPS1 is fed 85% by shallow ground water, while the source of BS5 is almost completely deep ground water [44]. Samples from all pools were observed to have low biomass. The organisms that were present were small microbes; no large microbes were observed. The approximate concentrations of motile organisms per field of view for GPS1 and BS5 were < 1 and 4, respectively (S18, S19 Video). The trajectory of select organisms from GPS1 samples is shown in Fig 11A. This image is a composite rendered by the minimum pixel intensity projection of multiple z-planes; the projected images show the organism's trajectory over 7 s. An intensity reconstruction image from the BS5 sample showed morphology of an individual cell (Fig 11B). Morphologies of small microbes generally were spherical and ~ 1 μm in diameter. Swimming patterns with reversal events were observed and tracked; two of these tracks can be seen in Fig 11C. Ground-truthing by DNA sequencing indicated concentrations of cells in GPS1 and BS5 were < $1.0 \times 10^1$ to $3.7 \ [\pm 0.5] \times 10^3$ cells $mL^{-1}$, respectively [77].

### 3.7 Cryopeg – Utqiaġvik, AK

Cryopeg brine was imaged directly after sampling in the permafrost tunnel at –6°C, but no evidence of active motility was seen in the 18 datasets that were recorded. Each dataset consisted of a total of 1 min at 15 fps. Many had particles that were consistent with small microbial shapes, but this possible biosignature was difficult to discern from mineral or other debris. The remaining sample was returned to the field laboratory in order to attempt to induce motility using temperature and salinity gradients. Cryopeg brine was then subjected to linear temperature gradients from 7.5°C to 32.0°C for approximately one hour. Discrete sections of the sample chamber were imaged corresponding to temperatures of 7.5°C, 17.5°C, and 32.0°C. Small microbial motility was observed at all of these temperatures. The concentrations of motile organisms observed at 7.5°C, 17.5°C, and 32°C were roughly $1 \times 10^4$, $2 \times 10^5$, and $4 \times 10^4$ motile small microbes/mL, respectively. These motile small microbes were difficult to identify and track due to their low contrast (Fig 12A). All of the tracked motile organisms were slow moving with an average speed of 6 μm/s, N = 3 (5 μm/s, 7 μm/s, 7 μm/s). Even at these

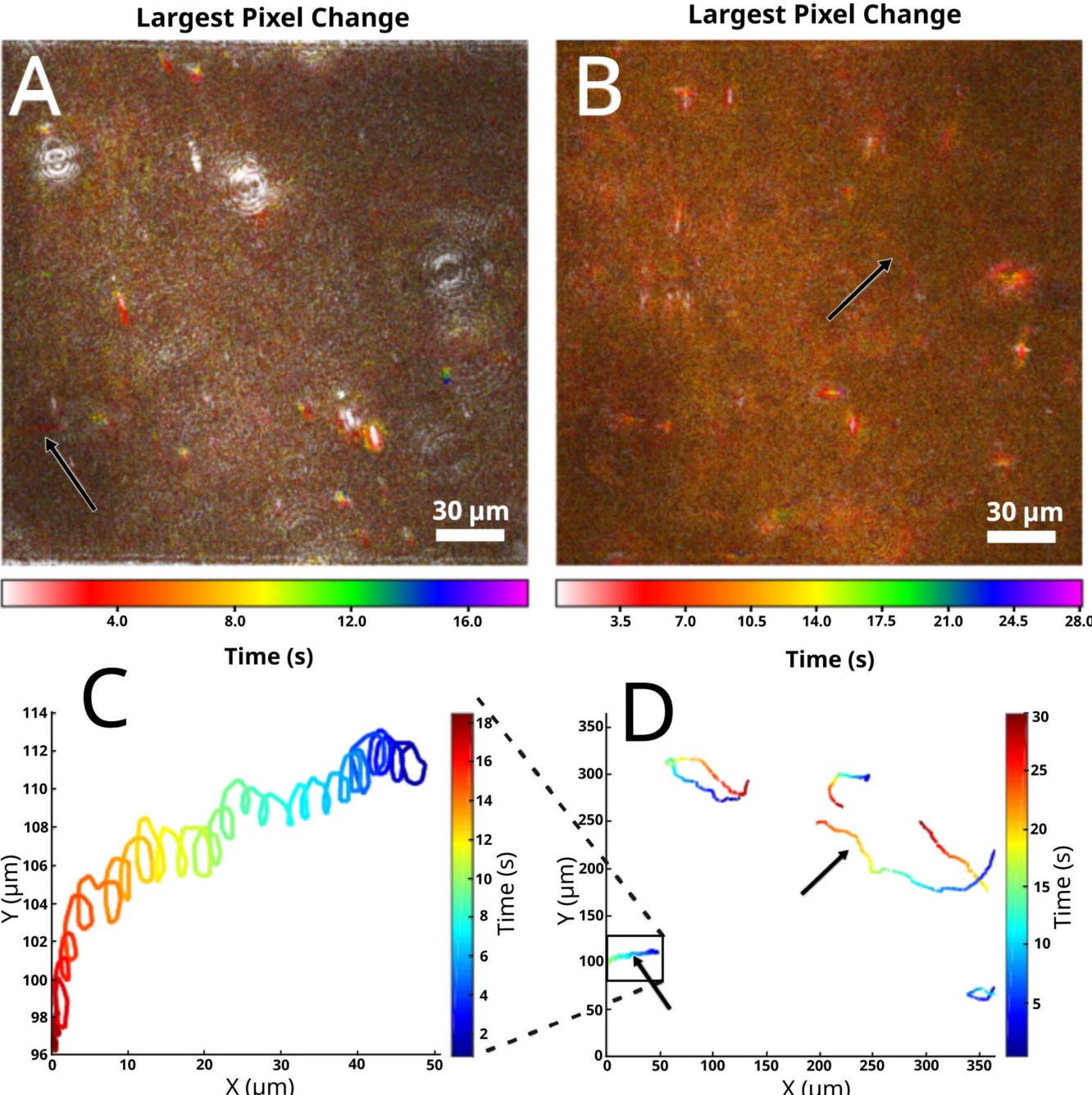

**Fig 10. Motility from Mt. St.** Helens glacial cave ice samples. (A, B) MHI images showing motion of a single recording conducted *in situ* immediately after ice melted. All motile small microbes in glacial ice samples had low SNR and were therefore difficult to identify in MHI. Arrows in each image indicate an observable trajectory. Both MHIs were created using reconstructed amplitude images where each image in the time series was the z-plane where the microbe of interest was in focus. This approach allowed the MHI to detect these two trajectories. (C) MATLAB plot showing the corkscrew trajectory of the motile small microbe shown in the bottom left of (A), (D). (D) MATLAB plot showing manually tracked 2D trajectories of all six motile small microbes observed in this recording.

speeds their motion was easily distinguished from the displacement that is caused by Brownian motion. Fig 12B shows manually tracked trajectories of motile organisms from the 17.5°C recording; one of these motile organisms can be seen in S20 Video. A previous report using

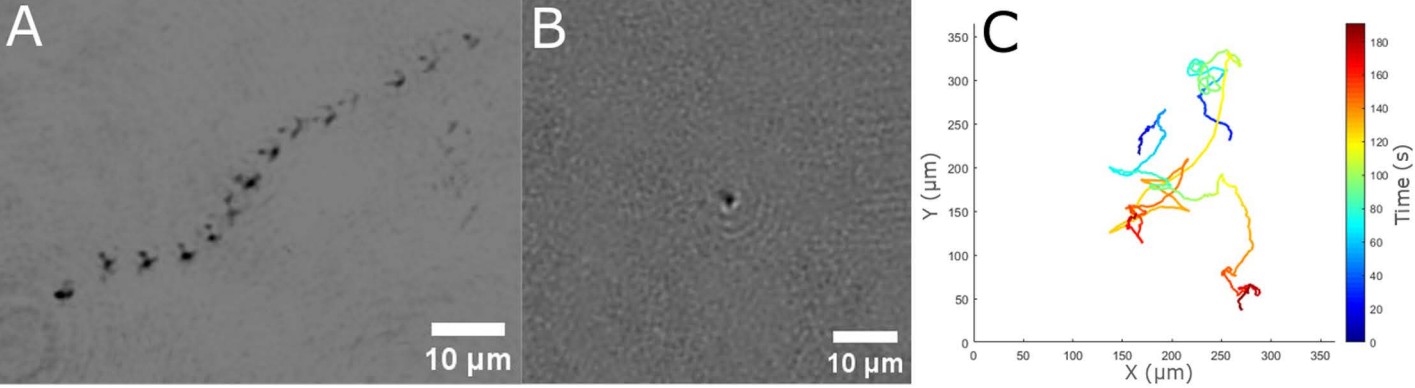

**Fig 11. Motility and morphology from The Cedars site.** (A) Selected trajectory of a motile microorganism found in the direct GPS1 spring sample. This image is a composite image rendered by the minimum pixel intensity projection of 7 s of data. (B) A single plane intensity reconstruction of a select micrometer-sized organism from the BS5 spring sample. (C) MATLAB 2D plot of two tracked motile small microbes from the BS5 pool.

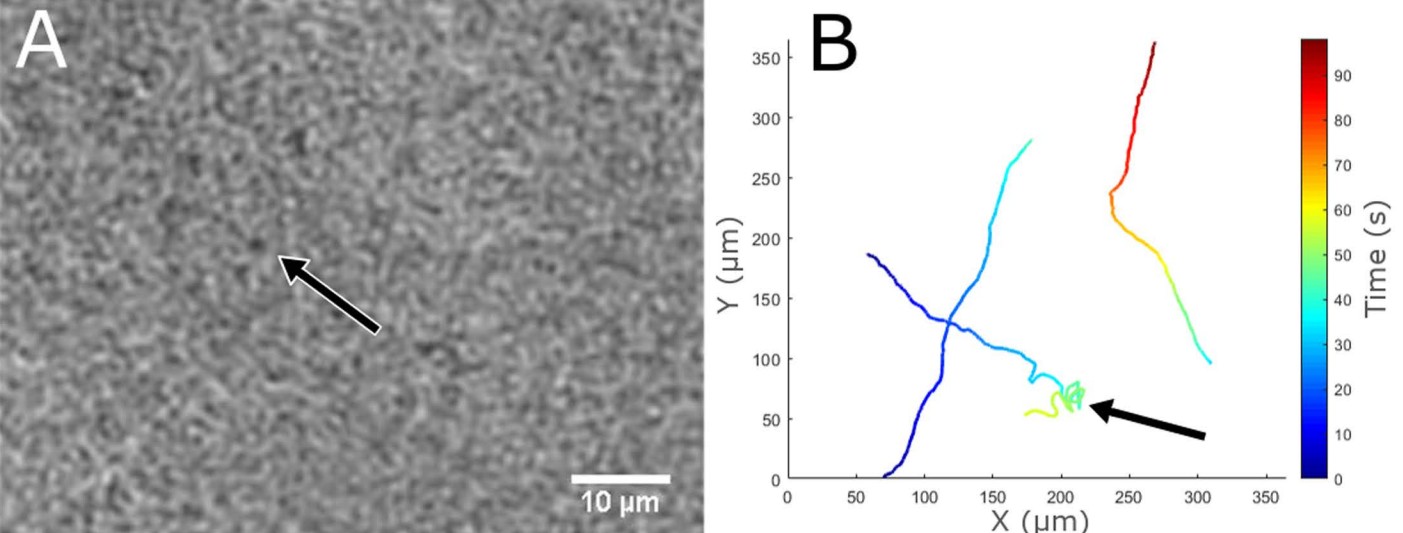

**Fig 12. Morphology and motility biosignatures from cryopeg brine samples.** (A) A single plane intensity reconstruction of a select micrometer-sized organism in a brine sample at 17.5°C. The arrows in (A) and (B) are identifying the same motile low contrast small microbe from the same recording. (B) A MATLAB 2D plot of small microbial motile organisms, showing trajectories of motile organisms at 17.5°C in a cryopeg brine sample from a thermal gradient experiment.

fluorescent staining identified a cell concentration in this same cryopeg brine of $1.4 \times 10^7$ cells mL$^{-1}$, and DNA sequencing indicated only prokaryotes in these samples [75]. Another report using DNA sequencing identified the presence of genes for flagellar assembly [78].

Cryopeg brine samples were subjected to a salinity gradient by a point source of low salinity solution. A needle tip filled with low salinity (3.5%) sterile agarose was used to establish and maintain a salinity gradient within the field of view of the DHM while imaging took place. No change was observed in the motility characteristics of the cryopeg brine sample. Time-lapse data also showed no net aggregation or dispersion of biomass towards or away from the salinity gradient, respectively.

An overview of observed biosignatures from samples is given in Table 2.

**Table 2. Summary of data analysis: sample types and motility observed.**

| Site | Sample type | Spontaneous motility | | Heat-stimulated motility (temperature) | | Nutrient-stimulated motility (stimulant) | Chemotaxis (stimulant) | Morphological biosignature |
|---|---|---|---|---|---|---|---|---|
| | | Large microbe | Small microbe | Large microbe | Small microbe | Small microbe | Small microbe | |
| Newport Beach, CA | Seawater | +++ | + | Not done | Not done | Not done | Not done | Diatoms, ciliates, flagellates, filaments |
| Malene Bay, Greenland | Seawater | ++ | + | ++ (−4°C, 4°C) | ++ (−4°C, 4°C) | +++ (2216, serine) | yes (serine, aspartate) n (glucose, trehalose) | Diatoms, flagellates |
| Malene Bay, Greenland | Sea ice brine | + | − | + | + (4°C) | ++ (2216, serine) - (trehalose) | yes (serine) n (glucose, trehalose) | Diatoms, flagellates |
| Salton sea | Hot saline spring | − | + | +++ (30°C) | +++ (30°C) | Not done | yes (serine) | Diatoms |
| Utqiaġvik, AK | Seawater | + | + | Not done | Not done | Not done | Not done | ND |
| Utqiaġvik, AK | Sea ice brine | + | + | Not done | Not done | Not done | Not done | ND |
| Utqiaġvik, AK | Cryopeg brine | − | − | − | +, ++, + (7.5°C, 17.5°C, 32°C) | n | no | ND |
| Badwater Basin, NV | Hypersaline pool | + | +++ | Not done | Not done | Not done | no | Benthic mats, diatoms |
| Ash Meadows, NV | Freshwater spring | + | +++ | - (30°C) | +++(30°C) | Not done | no | Ciliates |
| The Cedars, CA | BS5 Hyperalkaline, low-sodium pool | − | + | Not done | Not done | Not done | Not done | No large microbes, featureless low-index cells only |
| The Cedars, CA | GPS1 Hyperalkaline, low-sodium pool | − | + | Not done | Not done | Not done | Not done | No large microbes, featureless low-index cells only |
| Mt. St. Helens, WA | Glacier discharge | − | + | Not done | Not done | Not done | Not done | No large microbes, featureless low-index cells only |
| Mt. St. Helens, WA | Hot spring/glacier mix, midstream | +++ | +++ | Not done | Not done | Not done | Not done | Benthic mats, filaments, ciliates, flagellates |
| Mt. St. Helens, WA | Hot spring/ glacier mix, downstream | +++ | +++ | Not done | Not done | Not done | Not done | Flagellates, ciliates |
| Mt. St. Helens, WA | Glacier cave | − | + | Not done | Not done | Not done | Not done | No large microbes, featureless low-index cells only |

+++= highly abundant, multiple motile cells in each frame;

++= abundant, at least one motile cell per frame;

+ = present, but not in all frames; - = absent (cells not seen); ND = was not determined.

## 4. Discussion

As indicated in a 2016 NASA report, microscope-based identification of cell-like objects on future missions to other planetary bodies, whether or not the objects are definitively alive, will assist other means of characterization such as mass spectrometry, Raman or infrared spectroscopy, or other chemical analyses [3]. Chemical signals coming from a known self-contained object, or a selection of such objects, will be more compelling as biosignatures when images showing the objects accompany the chemical data contain biosignatures of their own. This study gives context on how to best use a DHM instrument to detect biosignatures in extreme environments. The *in situ* identification of at least one biosignature (i.e., motion, morphology, or optical properties) was confirmed at every site. Even though not all biosignatures were

present in each recording these results help build a case and framework for how to search for these biosignatures during future extant life detection missions.

Label-free amplitude and phase video microscopy can provide an array of biosignatures, some definitive and others suggestive. Either active motility or clear morphological features, if observed on another planet, would be a compelling biosignature, especially in combination with chemical studies demonstrating the presence of complex molecules and/or disequilibrium. Absorbance and refractive index optical properties are not definitive biosignatures on their own but fall into this suggestive category. The prevalence of motility, morphology, and optical property biosignatures in analog sites of astrobiological significance has remained largely unexplored. In this study, we observed motility and optical property biosignatures in samples from all the environments tested. Morphological features were identified in all environments except for cryopeg, alkaline spring pool, and subglacial void. The slowest microbes moved well above distinguishable speeds from Brownian and other forms of abiotic motion. Simple stimuli resulted in dramatic increases in motility in seawater and sea ice brines. Raising the temperature of sea ice brines from ambient (< 0ºC) to 4ºC resulted in small microbial motility, as did addition of 1 mM L-serine (but not other amino acids or glucose). Gradients of L-serine were not required to observe motility, though they could be used for further confirmation of directed swimming. Larger, clearly eukaryotic organisms were spontaneously motile in all sea ice and brine samples.

In samples from hot environments, cooling of the samples to room temperature led to loss or reduction of motility, and re-warming to motility. These results underscore the importance of temperature and indicate that the ambient temperature at which the cells are collected or observed in the laboratory may not be their optimum. The ability to ramp temperatures up and down would be crucial for any mission design. This stimulus is non-Earth-centric and straightforward to implement in a flight instrument.

Understanding motility *in situ* can bring insight into fundamental microbiological processes, oceanic nutrient cycling [79,80], and the origin and evolution of early life on Earth [81]. Motility is regulated by expression of flagellar-related genes and by the operation of the flagellar motor. Eukaryotic and archaeal flagellar function are dependent upon ATP [82,83], while bacteria can use an ion gradient across the cell membrane to drive flagellar function, usually a proton gradient [84,85]. In hyperalkaline environments, where the high pH creates a drastic background proton gradient across the cell membrane, a sodium gradient can substitute for the proton gradient for generation of chemiosmotic energy for flagellar motility [86]. Mechanisms by which organisms swim in both high pH and low sodium environments remain unclear.

Other types of stimuli may also need to be applied in samples where motility is not immediately evident. L-serine was shown to be a potent stimulus of motility in marine environments [87]. Panels of L- and D- amino acids and sugars might be used in a flight instrument to water worlds in our solar system, where the biochemistry could be assumed to be like Earth's [88,89]. In an extraterrestrial environmental sample, simply heating it to lysis (of cells as we know them) and then mixing it with a fresh sample may provide appropriate stimulants to the second sample. Further research into this suggested *in situ* stimulant technique would be an important avenue of research before incorporated into any life detection mission.

Developing panels of useful stimuli will require further study and understanding of the physicochemical and physiological drivers of microbial motility. The rapid induction of motility by stimuli indicates that the organisms present expressed all of the necessary genes for flagellar motility, but were not capable of swimming under their *in situ* conditions. Loss of motility may result from lack of energy sources, loss of necessary ionic gradients, or inhibitors. Advances in volumetric and video microscopy [90] have begun to reveal the importance of

microbial motility across a wide range of temporal and spatial scales [91–95]. Motility plays a role in environments so oligotrophic that swimming has been predicted to be a less favorable trait (e.g., ocean gyres) [87,96], as well as those so eutrophic that nutrient limitation is not a factor (e.g., the gut) [97,98]. Even though motility may not be as favorable in these environments, on the microscale at which microbes operate, there are gradients to drive taxis.

The benthic mats at the Mt. St. Helens midstream site illustrate the importance of multi-scale, multi-modal imaging in life detection. Where the hot spring and glacier waters mixed and the temperature was 35-37ºC, biosignatures were tens of centimeters to meters in size, representing thick, ropy mats of colors distinct from the surrounding rock. Microscopy at low magnification was sufficient to illustrate the cellular nature of the mats. Both autofluorescence and motility were abundant and striking. However, without intelligently driven site selection based upon context images, the mats might be missed. The water surrounding the mats was very low in biomass. A few meters away, in pure glacier run off near 0ºC, biomass was also very low. It is highly likely that extraterrestrial life will also show peaks of distribution around energy sources, and so picking where to look on the kilometer to meter and even centimeter scale is every bit as important as microscopy.

This study highlights the importance of ground truthing to complement DHM data. Fluorescence imaging and genetic sequencing can provide crucial information to confirm imaging biosignatures. Both of these techniques are being developed for space [12,99–101], though both rely on biosignatures that may be Earth-centric and present some significant barriers to implementation on other worlds.

Two DHM instruments were used in the field: one, the original field deployable instrument described in 2016 [27]; and the other, a smaller version with nearly identical performance characteristics. This work not only shows the technique of DHM as a suitable modality for life detection, but that the physical architecture of DHM is robust enough to withstand a multi-year deployment campaign to various harsh environments without any catastrophic failures of any kind. We have recently reported a DHM coupled to a lightfield microscope for multi-modal DHM and fluorescence [102]. The addition of fluorescence microscopy would permit the detection of additional biosignatures due to autofluorescence or specific dyes that target biosignature molecules [101]. These studies, including the current one, represent the first steps in an emerging field that will allow for rational designs of missions to use microscopy to help detect extant microorganisms *in situ* in lander or flyby missions.

## Supporting information

**S1 Fig. Sampling Sea ice and cryopegs in Utqiaġvik, AK.** (A) Removing snow to access sea ice for coring and brine sampling. (B) Sea ice core. (C) Diagram of tunnel allowing access to cryopegs. (D) Photo of access tunnel.
(TIF)

**S2 Fig. Kings Pool in Ash Meadows, NV.** Within the boundary of Ash Meadows National Wildlife Refuge exists many spring fed pools, imaged here is Kings Pool.
(TIF)

**S3 Fig. Context photos from The Cedars, CA.** (A) Shows the collecting of DHM data from BS5 using the SHAMU instrument. (B) A photo of GPS and Manuel Bedrossian preparing tubing for sample collection.
(TIF)

**S4 Fig. Sites from the Mt. St. Helens crater.** The sites reported here are glacier discharge (upstream), the site of mixing of glacier discharge with hot spring (Midstream or Mixed (up)),

and mixed downstream. Basemap derived from USGS Data Series 904 authors by Adam Mosbrucker [103].
(TIF)

**S5 Fig. Map of the Mt. St. Helens Mothra cave site.** This map was derived during the fieldwork within the Mothra cave. The top depiction of the glaciovolcanic cave is from above looking down towards the center of the Earth. The bottom depiction shows the profile or cross section of the cave. The orientation of the profile image corresponds to the top depiction of the cave. Republished from [40], Image by Christian Stenner licensed under CC-BY 4.0.
(TIF)

**S6 Fig. Field deployable DHM instruments.** (A, B) Optical train common to both designs. (C, D) Original field design, called SHAMU, total weight 10 kg (base images A-D previously published in [27]). (E, F) Compact folded field design, Son of SHAMU, total weight 6 kg. (G) Son of SHAMU in the Mt. St. Helens crater. (H) SHAMU in Death Valley.
(TIF)

**S7 Fig. *In situ* results from sea water and sea ice brine samples collected in Barrow, AK.** (A) Select trajectory of a motile microorganism found in the sea ice brine sample. This image is a composite image rendered by the minimum pixel intensity projection of 5 seconds of data. (B) Select trajectory of a motile microorganism found in the ice/ocean interface sample. This image is a composite image rendered by the minimum pixel intensity projection of 5 seconds of data. (C) A single plane intensity reconstruction showing a select microorganism observed at the ice/ocean interface sample. (D) Trajectories of the heterogeneous *in situ* sample taken from the ice/ocean interface. This plot shows trajectories of multiple organisms within the field of view of the DHM instrument and is color coded with respect to time.
(TIF)

**S8 Fig. Ground-truth imaging of Salton Sea sample.** (A) Low-power image of Salton Sea sample stained with SYTO9, under phase contrast (left) and fluorescence (right). (B) High-power image of a single large microbe stained with SYTO9, under phase contrast (left) and fluorescence (right). (C) Unstained Salton Sea large microbes showing autofluorescence upon green excitation, under phase contrast (left) and fluorescence (right). (D) Salton Sea sample stained with SYTO9 and demonstrating red autofluorescence, showing the fusion of the green and red channels.
(TIF)

**S9 Fig. Fluorescence imaging of benthic mats from Mt. St. Helens midstream.** (A) Low power image of mat chlorophyll autofluorescence. The dark areas are minerals. (B) Higher power image showing segmented photosynthetic cells. The inset shows SYTO9 staining of a round photosynthetic cell.
(TIF)

**S10 Fig. Mt St. Helens crater downstream site biosignatures.** (A) Amplitude reconstruction of a single plane showing several 2-5 μm sized cells. (B) MHI of recording showing rapid swimming of 5 different cells within the 30 s time frame of the video. (C) Tracks of cells in (B). Red arrows are pointing to tracks of drifting cells that would be ideal for using to apply drift subtraction of motile cells.
(TIF)

**S1 Video. *In situ* seawater sample video Utqiaġvik, AK.** Motile small microbial cells are seen in this seawater sample video.
(AVI)

**S2 Video. *In situ* sea ice brine sample video Utqiaġvik, AK.** Motile large and small microbial cells are seen in this video.
(AVI)

**S3 Video. *In situ* seawater sample video Malene Bay, GL.** Motile small microbial cells are seen in this video.
(AVI)

**S4 Video. *In situ* seawater sample video Newport Beach, CA.** Motile largeand many small microbial cells are seen in this video.
(AVI)

**S5 Video. *In situ* video Nuuk, GL.** Spontaneous motility of 1 large microbe observed in Nuuk Greenland sample.
(AVI)

**S6 Video. Thermal control video Salton Sea, CA 22ºC.** No motility observed.
(AVI)

**S7 Video. Thermal experiment video Salton Sea, CA 25ºC.** Motility of organisms heated to 25ºC from Salton Sea samples.
(AVI)

**S8 Video. Thermal experiment video Salton Sea, CA 30ºC.** Motility of organisms heated to 30ºC from Salton Sea samples.
(AVI)

**S9 Video. Thermal experiment video Salton Sea, CA 42ºC.** Slow large microbe grazing like movement observed at 42ºC.
(AVI)

**S10 Video. Chemotaxis video Salton Sea, CA.** Chemotaxis towards serine was observed in time lapse, 1 frame every 30 sec. Cloud formation of small microbes appeared after 5 mins, and it appears likely that large microbes are preying on small microbes drawn towards the serine gradient.
(AVI)

**S11 Video. *In situ* video Badwater Basin, Death Valley, CA.** Spontaneous motility of 4 small microbes, indicated by the red arrows, and 1 large microbe.
(AVI)

**S12 Video. I*n situ* video Badwater Basin, Death Valley, CA.** One of the two motile organisms tracked with motility pattern resembling run-reverse, location indicated by the red arrow. To reduce video data submitted, only every other frame of original recording is included and so the recording is running at 7.5 fps.
(AVI)

**S13 Video. *In situ* video Kings pool, Ash Meadows, NV.** An *in situ* video from Kings pool in Ash Meadows National Wildlife Refuge showing two motile small microbes, indicated by the red arrows. Every other frame of original recording is included and so the recording is running at 7.5 fps.
(AVI)

**S14 Video. *In situ* video Mt. St. Helens glacial runoff, WA.** Video showing some motile organisms as well as microminerals. Some suspended non-motile particles with significantly are seen with higher contrast than motile organisms.
(AVI)

**S15 Video.** *In situ* **video Mt. St. Helens midstream site, WA.** A video recording of the orange deposit sample shows motile small microbe, indicated by red arrows, that is clearly different in refractive index from the surrounding microminerals. Every other frame of original recording is included and so the recording is running at 7.5 fps.
(AVI)

**S16 Video.** *In situ* **video Mt. St. Helens midstream site, WA.** Video shows a high density of motile organisms swimming outside the plane of the green mat.
(AVI)

**S17 Video.** *In situ* **video Mt. St.** Helens subglacial void, WA. Recording from ice samples that were melted under ambient temperatures and immediately recorded. Three motile small microbes can be seen within the red circles that help indicate their location.
(AVI)

**S18 Video.** *In situ* **video Barnes Spring 5, The Cedars, CA.** Video showing one of the motile microbes identified in BS5, location indicated by the red arrow. Every other frame of original recording is included and so the recording is running at 7.5 fps.
(AVI)

**S19 Video.** *In situ* **video Grotto Spring Pool, The Cedars, CA.** Video showing motility of one of the motile organisms in the GPS recordings.
(AVI)

**S20 Video. Thermal experiment video Cryopeg – Utqiaġvik, AK 17.5ºC.** Video showing the motion of one of the motile organisms identified during the thermal gradient experiment at 17.5 ºC. The trajectory seems to follow a run-reverse pattern.
(AVI)

## Acknowledgements

We thank collaborators J. Kent Wallace, Eugene Serabyn, and Kurt Liewer at Jet Propulsion Laboratory for the design and development of our digital holographic microscope system. Additionally, we thank Santos F. Fregoso of the Jet Propulsion Laboratory for development of DHMxSuite software package that was used for acquisition, processing, and analysis of DHM recordings. We thank Gordon Max Showalter for his collaboration and work in the field in the Barrow Permafrost Tunnel in Alaska and in Malene Bay in Greenland. We thank Eddy Cartaya and Andreas Pflitsch for organizing and leading the 2021 Mt. St. Helens glacier cave field work. We thank Mark Skidmore, Peter Doran, and the rest of the THOR team for organizing and assisting in the 2020 Mt. St. Helens crater field work. We thank the Mount Saint Helens Institute for allowing us access to the field work campsites in the 2020 Mt. St. Helens field work. We thank Henry Sun and Brian Hedlund for organizing and leading field work in Death Valley and Ash Meadows.

## Author contributions

**Conceptualization:** Manuel Bedrossian, Jody W. Deming, Chris A. Lindensmith, Jay L. Nadeau.

**Data curation:** Carl Snyder, Manuel Bedrossian, Casey Barr, Jay L. Nadeau.

**Formal analysis:** Carl Snyder, Manuel Bedrossian, Casey Barr, Jay L. Nadeau.

**Funding acquisition:** Jody W. Deming, Chris A. Lindensmith, Jay L. Nadeau.

**Investigation:** Carl Snyder, Manuel Bedrossian, Casey Barr, Jody W. Deming, Chris A. Lindensmith, Christian Stenner, Jay L. Nadeau.

**Methodology:** Carl Snyder, Manuel Bedrossian, Casey Barr, Jody W. Deming, Jay L. Nadeau.

**Project administration:** Jody W. Deming, Chris A. Lindensmith, Jay L. Nadeau.

**Resources:** Jody W. Deming, Chris A. Lindensmith, Jay L. Nadeau.

**Supervision:** Jody W. Deming, Christian Stenner, Jay L. Nadeau.

**Validation:** Carl Snyder, Manuel Bedrossian.

**Visualization:** Carl Snyder, Manuel Bedrossian, Jay L. Nadeau.

**Writing – original draft:** Carl Snyder, Manuel Bedrossian, Casey Barr, Jay L. Nadeau.

**Writing – review & editing:** Carl Snyder, Manuel Bedrossian, Casey Barr, Jody W. Deming, Chris A. Lindensmith, Christian Stenner, Jay L. Nadeau.

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
