## [Decision Letter · Decision Letter 0]

2 Nov 2023

PONE-D-23-22462Extant Life Detection Using Label-Free Video Microscopy in Analog Aquatic EnvironmentsPLOS ONE

Dear Dr. Snyder,

Thank you for submitting your manuscript to PLOS ONE. After careful consideration, we feel that it has merit but does not fully meet PLOS ONE’s publication criteria as it currently stands. Therefore, we invite you to submit a revised version of the manuscript that addresses the points raised during the review process.

We look forward to receiving your revised manuscript.

Kind regards,

Michael Schubert

Academic Editor

PLOS ONE

Journal Requirements: 

"Funding was provided by the National Science Foundation (1828793) and a National Aeronautics and Space Administration (NASA) PSTAR grant (80NSSC18K1738) for THOR (Thermal High-voltage Ocean-penetrator Research platform).  Portions of this work were supported under a contract from, or performed at, the Jet Propulsion Laboratory, California Institute of Technology, under a contract with the National Aeronautics and Space Administration. Other portions were supported by grant ARC-1203267 from NSF and grant 5488 from the Gordon and Betty Moore Foundation (JWD)"

7. We note that Figure 1 & Figure S5 in your submission contain map images which may be copyrighted. All PLOS content is published under the Creative Commons Attribution License (CC BY 4.0), which means that the manuscript, images, and Supporting Information files will be freely available online, and any third party is permitted to access, download, copy, distribute, and use these materials in any way, even commercially, with proper attribution. For these reasons, we cannot publish previously copyrighted maps or satellite images created using proprietary data, such as Google software (Google Maps, Street View, and Earth). For more information, see our copyright guidelines: http://journals.plos.org/plosone/s/licenses-and-copyright.

(1) You may seek permission from the original copyright holder of Figure 1 & Figure S5 to publish the content specifically under the CC BY 4.0 license.  

8. We note that Figure S3 includes an image of a participant in the study. 

As per the PLOS ONE policy (http://journals.plos.org/plosone/s/submission-guidelines#loc-human-subjects-research) on papers that include identifying, or potentially identifying, information, the individual(s) or parent(s)/guardian(s) must be informed of the terms of the PLOS open-access (CC-BY) license and provide specific permission for publication of these details under the terms of this license. Please download the Consent Form for Publication in a PLOS Journal (http://journals.plos.org/plosone/s/file?id=8ce6/plos-consent-form-english.pdf). The signed consent form should not be submitted with the manuscript, but should be securely filed in the individual's case notes. 

Please amend the methods section and ethics statement of the manuscript to explicitly state that the patient/participant has provided consent for publication: “The individual in this manuscript has given written informed consent (as outlined in PLOS consent form) to publish these case details”. 

Reviewers' comments:

Reviewer's Responses to Questions

**Comments to the Author**

1. Is the manuscript technically sound, and do the data support the conclusions?

Reviewer #1: Partly

Reviewer #2: Partly

Reviewer #3: Partly

Reviewer #4: Yes

2. Has the statistical analysis been performed appropriately and rigorously? 

Reviewer #1: N/A

Reviewer #2: No

Reviewer #3: No

Reviewer #4: N/A

3. Have the authors made all data underlying the findings in their manuscript fully available?

Reviewer #1: Yes

Reviewer #2: Yes

Reviewer #3: Yes

Reviewer #4: Yes

4. Is the manuscript presented in an intelligible fashion and written in standard English?

Reviewer #1: Yes

Reviewer #2: Yes

Reviewer #3: Yes

Reviewer #4: Yes

5. Review Comments to the Author

Reviewer #1: Your work is very interesting and the effort addressing motility as a biosignature makes sense considering our experience in trying to evaluate extant prokaryotes in desert environments.

Some issues:

- How do you differentiate between eukaryotic and prokaryotic cells? Is the size of the "cells" the only difference?

- It is confusing calling the sample "desert spring system" (L 291) considering that, following Lee et al 2018, hypersaline pools are "thermodinamically moderate", and "contains all domains of life and perform complete biogeochemical cycling". Being a hot spring defines it better than being located in a desert.

Specific comments:

Fig 1. L 116 Please change second by thrisd column.

L 253 Figure 3 does not have E and F.

L 329 How do you confirm predation?

L 451 and others Should it says selected trajectory?

Reviewer #2: Summary

The authors present results using holographic microscope in situ from a collection of various environments known to be extreme for life (ice brines, hypersaline pools,etc…), based on prototypes (one of them have been previously published). The authors also present stimuli to interfere with cell motility. Such results could have some useful information for astrobiology to justify to use of holography microscopy on a mission.

However, if they provide a catalog of various results in term of biological environment, taking into account on the difficulties to get such results in situ, still, the number of sample per site is limited which makes difficult to get a statistically strong measurement.

Author provide an overview of biosignature they obtained, but unfortunately many are not quantified even when it is abundant. Such quantification, with a homogenous method and sufficient statistics would really help providing significant biosignature.

Example and evidences

Major issues

Some results have already published : fig 3 was present in Fig 6 in their paper (26) , Fig S6 was Fig1 for instance.

Careful relecture is necessary. For instance, 1) some figures doesn’t correspond: in Figure 3 D to F subpanels are absent. 2) in the text, mismatch between Fig S7 and S8. 3) for Badwater Basin, “motile cells” is used but not clear if it concerns eukaryotes or prokaryotes.

In some movies, we can see global drift. It is not clear how it has been taken into account

For some results, (in different sections, only 2 or 3 cells have been measured), which is really too low. Moreover, it makes difficult to consider a Standard Deviation.

Minor issues

Using time in seconds instead of frame number would help for HMI. For the same reason, adding a timestamp in movies helps to get an idea of the timescale.

Movies contrast is variable, which makes difficult to evaluate. Having all the data with same contrast would be useful

Reviewer #3: This manuscript describes a search for motile microorganisms in various environments, with the aim of understanding how to design astrobiological survey missions. The introduction and discussion are very well written and clear, but the methods/materials section and results section should be revised.

Major points:

1. I was confused about the relationship between HELM, the reconstruction method, and the tracking algorithm. Are the results from the reconstruction sent to the tracking module, and these results sent to HELM? A flowchart would be useful here.

2. The biggest question I have about any detection algorithm is how sensitive the results are to the parameters chosen for the algorithm. There are likely many parameters and detection thresholds in the localization, tracking, and motility detection steps. These should be specified, and there should also be a section that explains and justifies how the parameters/thresholds are chosen. Also there should be some analysis of how sensitive the results (for example, the cell counts and the number of detected motile organisms) are to the choices of parameters and thresholds.

3. The other question in any biological survey is what protocols are used to reduce the risk of contamination. Especially with the samples that show very few microorganisms, or no motility until heating, contamination is a concern. While the authors discuss the use of sterile supplies for sampling, they do not discuss how the imaging and heating environment is kept sterile. Detailed protocols on sterilization and cleaning procedures are needed for others to reproduce the work.

4. The results section was difficult for me to follow. The results are organized by field site, but the methods used are different from site to site. It seemed to me that some sites had no ground truthing experiments performed, or at least none described (Desert Spring System and Ice from Subglacial Void, for example). Some sites had no cell counts. And different detection methods for motility are presented for the sites. It would be more helpful for the reader if these results were organized in a more systematic way. Table 2 is helpful, but the text needs to be revised. Also, ground truth experiments need to be reported for each case.

5. The reported abundance of motile cells (table 2) should be statistically quantified and the uncertainties reported. Uncertainties related to three factors should all be considered: heterogeneity (sample-to-sample variation), sample size (number of detected organisms in field of view), and model uncertainty (variation in number of detected motile organisms with variation in detection thresholds). The total uncertainty is the convolution of these three sources. For these low concentrations, Poisson statistics is likely a good way to estimate uncertainty due to sample size. The model uncertainty should also be straightforward to estimate by varying the detection parameters and thresholds over a reasonable range (see point 2). Uncertainty due to heterogeneity requires quantifying sample-to-sample variation, which may not be available for all sites, but some estimate should be provided here.

Minor points:

6. More details are needed on how the fluorescence-based cell counting was done (lines 236-237).

7. Panels E and F are missing from Figure 3 in the version I received

8. For the "Glaciovolcanic: Glacial Melt and Hot Spring" section, it's not clear to me what ground-truth experiments were done to validate the classification of minerals versus organisms in the DHM analysis.

Reviewer #4: This is a very detailed study of microbial motility detection in a wide range of extreme environments. It is shown that Digital Holographic Microscopy (DHM) can be used in the field to effectively detect motile microorganisms, and thus provide a firm biosignature. This field-based technology is highly relevant for future planetary missions to search for life beyond Earth (particularly Ocean Worlds). Motility itself will of course greatly limit such a search for life, since not all microorganisms evolved to motile forms. However, as the authors point out, in most aquatic environments some fraction of the microbial cells will be motile. Spontaneous, active motility stimulated by e.g. a temperature change, addition of nutrients or a salinity gradient, would provide a very strong biogenicity test. The images produced by DHM were processed using the software package HELM to detect motility and subsequently motion history images (MHI's) were produced. These clearly showed the movement paths of various prokaryotes and eukaryotes. I think this is a valuable study showing a technology that will be of high interest future space missions.

The manuscript is well written and many figures and videos are shown to support the findings. This is good, but also quite confusing for a reader. It is often difficult to follow exactly what is seen in a video in the supplement and how this translates to an observation in a main figures in the paper. Also, it seems that some figures are missing/misplaced, or not referred correctly. Below I list some of these issues. I suggest publication after these minor revisions.

Comments:

line 253 and 264: Fig.3E and 3F are described here but these images are missing in Fig.3.

line 316-319: Reference is made to fluorescence microscopy images here in Fig.S7. Should this be Fig.S8?

line 336-346: In some cases it is difficult for a reader to see how the images link to the motions in the videos. For instance in Fig.5C,D the MHI is shown for motility in video S12. It is really difficult to see what exactly is being tracked. Is it possible to add arrows into the movie to point out what exactly is moving? This is just a suggestion to make it more clear.

line 355: glacial melt samples are presented with reference to Fig.S3. Should this actually be Fig.S4?

line 360-362 '... contained a large amount of silt and microminerals (Fig.6)'. These have elongated and spindle-like shapes. I really wonder what type of silt/minerals have that shape. Can the authors better clarify this? And can it be pointed out in Fig.6A?

6. PLOS authors have the option to publish the peer review history of their article (what does this mean? ). If published, this will include your full peer review and any attached files.

**Do you want your identity to be public for this peer review?** For information about this choice, including consent withdrawal, please see our Privacy Policy .

Reviewer #1: **Yes: ** Cecilia Demergasso

Reviewer #2: No

Reviewer #3: No

Reviewer #4: No

---

## [Author Response · Author response to Decision Letter 1]

2 May 2024

PLOS ONE Manuscript PONE-D-23-22462

Response to Reviewers

Dear PLOS ONE Editor,

Thank you for giving us the opportunity to submit a revised draft of the manuscript “Extant Life Detection Using Label-Free Video Microscopy in Analog Aquatic Environments” for publication in the PLOS ONE Journal. We recognize and greatly appreciate the time and labor that you all dedicated to providing input on improving our manuscript and are grateful for the insightful critiques on and valuable improvements to our paper. We have included many of the suggestions made. All changes within the manuscript are seen in red text. Please see below, in blue, for a point-by-point response to the reviewers’ comments and concerns. All page numbers refer to the revised manuscript file with tracked changes.

Editors’ Comments to the Authors and Respective Responses to Editor:

Author response: Thank you for sharing these style template documents. They were much easier to follow than what I found on the website. The PLOS ONE guidelines have been reviewed again and the manuscript has been revised to follow them to the best of our ability.

2.Please note that PLOS ONE has specific guidelines on code sharing for submissions in which author-generated code underpins the findings in the manuscript. In these cases, all author-generated code must be made available without restrictions upon publication of the work. Please review our guidelines at https://journals.plos.org/plosone/s/materials-and-software-sharing#loc-sharing-code and ensure that your code is shared in a way that follows best practice and facilitates reproducibility and reuse.

Author response: None of the programs used were generated by the authors. HELM is a software package developed by collaborators from JPL but none of the developers are on the paper. The TrackMate code used is open source and generated by other groups unaffiliated with us. Since none of the authors are the ones that generated the code I assume these guidelines on sharing the code do not apply.

3.Thank you for stating the following financial disclosure:

"Funding was provided by the National Science Foundation (1828793) and a National Aeronautics and Space Administration (NASA) PSTAR grant (80NSSC18K1738) for THOR (Thermal High-voltage Ocean-penetrator Research platform). Portions of this work were supported under a contract from, or performed at, the Jet Propulsion Laboratory, California Institute of Technology, under a contract with the National Aeronautics and Space Administration. Other portions were supported by grant ARC-1203267 from NSF and grant 5488 from the Gordon and Betty Moore Foundation (JWD)"

Author response: We have included the statement reading:

“... The funders had no role in study design, data collection and analysis, decision to publish, or preparation of the manuscript.”

4.In your Data Availability statement, you have not specified where the minimal data set underlying the results described in your manuscript can be found. PLOS defines a study's minimal data set as the underlying data used to reach the conclusions drawn in the manuscript and any additional data required to replicate the reported study findings in their entirety. All PLOS journals require that the minimal data set be made fully available. For more information about our data policy, please see http://journals.plos.org/plosone/s/data-availability.

Author response: I have updated both our Data Availability statement and cover letter to include a URL link to the dryad database that is storing our minimal dataset. I did not see that the minimal data set link should be included in the cover letter previously sorry for not including it before. The URL link is also provided here: https://datadryad.org/stash/share/uln9WKGPLlY0DqlsA7Q1Ia8JtbZ3uCPOopXhd9bjtMU

Author response: I misunderstood how to share the Dryad database. So the link I have provided should work for editors and reviewers moving forward. Dryad URL: https://datadryad.org/stash/share/uln9WKGPLlY0DqlsA7Q1Ia8JtbZ3uCPOopXhd9bjtMU

Author response: Thank you for catching the absence of our permission statement. Below is the added text to the Materical Methods section.

L160 – Samples were collected from 8 different locations totaling 15 different sample sites across North America. Excluding Salton Sea, permission to sample was obtained from all sites. Salton Sea sample location was at a public beach access. Permission to collect samples in Greenland was obtained from the Greenland Climate Research Center. The issuing authority for research near Utqiaġvik, AK, was UIC Science, LLC. Field work in the Mt. St. Helens crater was performed under a U.S. Department of Agriculture Forest Service temporary special-use permit. The Cedar springs are located on private land held in a trust and permission was granted by Roger Raiche and David McCrory. The Newport beach work was approved by Caltech’s Kerckhoff Marine Laboratory. Sample collection at Kings pool in Ash Meadows was under a special use permit granted by Ash Meadows National Wildlife Refuge. US National Park Service granted permission for sampling at the Badwater Basin Death Valley site.

7.We note that Figure 1 & Figure S5 in your submission contain map images which may be copyrighted. All PLOS content is published under the Creative Commons Attribution License (CC BY 4.0), which means that the manuscript, images, and Supporting Information files will be freely available online, and any third party is permitted to access, download, copy, distribute, and use these materials in any way, even commercially, with proper attribution. For these reasons, we cannot publish previously copyrighted maps or satellite images created using proprietary data, such as Google software (Google Maps, Street View, and Earth). For more information, see our copyright guidelines: http://journals.plos.org/plosone/s/licenses-and-copyright.

(1) You may seek permission from the original copyright holder of Figure 1 & Figure S5 to publish the content specifically under the CC BY 4.0 license.

Author response: Figure 1’s base image is from Wikimedia Commons and my understanding is that the base image is within the public domain and freely licensed content. Do I need to still request permission from a copyright holder? https://commons.wikimedia.org/wiki/File:North_America_laea_relief_location_map.jpg For this figure I have added the following text to the Fig 1 legend. I believe this should be sufficient but if it is not, please let me know.

L125 – “North America laea relief location map” by Uwe Dedering is licensed under CC BY-SA 3.0. https://commons.wikimedia.org/wiki/File:North_America_laea_relief_location_map.jpg

Figure S4 also has a map base. Its base map was produce in USGS data series 904 (https://pubs.usgs.gov/publication/ds904). I am not sure who to contact about getting copyright permission. In previous work that I have done, another author used this source to make a figure and they cited the USGS report and did not indicate copyright permission. I am altering the manuscript to do the same but I am not certain this is the correct method for doing this.

SI: L121 – Basemap derived from USGS Data Series 904 authors by Adam Mosbrucker [25]

SI: L318 – [25] Mosbrucker, A. R. (2014).High-resolution digital elevation model of Mount St. Helens crater and upper North Fork Toutle River basin, Washington, based on an airborne lidar survey of September 2009. Data Series 904 U.S. Geological Survey. https://doi.org/10.3133/ds904

Figure S5 is made by an author of the paper, Christian Stenner. Chris has given permission via the form linked above. I have altered the text to include the permission within the figure caption.

SI: L129 – The source of this map is author Christian Stenner. The map was a custom basemap image created author Christian Stenner under a Creative Commons Attribution License (CC BY 4.0).

8. We note that Figure S3 includes an image of a participant in the study.

As per the PLOS ONE policy (http://journals.plos.org/plosone/s/submission-guidelines#loc-human-subjects-research) on papers that include identifying, or potentially identifying, information, the individual(s) or parent(s)/guardian(s) must be informed of the terms of the PLOS open-access (CC-BY) license and provide specific permission for publication of these details under the terms of this license. Please download the Consent Form for Publication in a PLOS Journal (http://journals.plos.org/plosone/s/file?id=8ce6/plos-consent-form-english.pdf). The signed consent form should not be submitted with the manuscript, but should be securely filed in the individual's case notes.

Please amend the methods section and ethics statement of the manuscript to explicitly state that the patient/participant has provided consent for publication: “The individual in this manuscript has given written informed consent (as outlined in PLOS consent form) to publish these case details”.

Author response: The individuals imaged in S3 Fig are authors Manuel Bedrossian and Casey Barr. Figures 2, S1, S2, S3, and S6 all have images showing pictures of authors. Since you did not mention these other figures, I am assuming that S3 Fig is an issue because it is the only one that gives a name of the individuals in the image. I have removed the names from the legend. If we need to get consent forms from all authors shown in the images, please let me know and I will do that. The legend now reads:

SI: L90 – A photo showing tubing being prepared for sample collection at GPS.

Reviewers' Comments to the Authors and Respective Responses to Reviewers:

Reviewer 1

Your work is very interesting and the effort addressing motility as a biosignature makes sense considering our experience in trying to evaluate extant prokaryotes in desert environments.

Author response: Thank you for your kind words and suggestions on how to best improve our manuscript.

1. How do you differentiate between eukaryotic and prokaryotic cells? Is the size of the cell the only difference?

Author response: We distinguished eukaryotes and prokaryotes via size differential and the observation of organelles. Microbes classified as prokaryotes were consistently well below 5 mm and eukaryotes were above 20 µm in maximum dimension.

The revised text reads as follows:

L280 – We distinguished eukaryotes and prokaryotes via size differential and the observation of organelles. Microbes classified as prokaryotes were consistently below 5 µm and eukaryotes were above 20 µm in maximum dimension.

2. L291 — It is confusing to calling the sample "desert spring system considering that, the following Lee et al 2018, hypersaline pools are "thermodynamically moderate", and "contains all domains of life and perform complete biogeochemical cycling". Being a hot spring defines it better than being located in a desert.

Author response: We agree that the Salton Sea

---

## [Decision Letter · Decision Letter 1]

3 Jun 2024

PONE-D-23-22462R1Extant life detection using label-free video microscopy in analog aquatic environmentsPLOS ONE

Dear Dr. Snyder,

Thank you for submitting your manuscript to PLOS ONE. After careful consideration, we feel that it has merit but does not fully meet PLOS ONE’s publication criteria as it currently stands. Therefore, we invite you to submit a revised version of the manuscript that addresses the points raised during the review process.  Specifically, the reviewers highlight significant shortcomings concerning the reproducibility and the statistical significance of the experiments, suggesting that the claims made in the manuscript are not supported by the reported data. Unless the technical rigor of the study is improved, it will thus be impossible to recommend publication in PLOS ONE.

We look forward to receiving your revised manuscript.

Kind regards,

Michael Schubert

Academic Editor

PLOS ONE

Reviewers' comments:

Reviewer's Responses to Questions

**Comments to the Author**

1. If the authors have adequately addressed your comments raised in a previous round of review and you feel that this manuscript is now acceptable for publication, you may indicate that here to bypass the “Comments to the Author” section, enter your conflict of interest statement in the “Confidential to Editor” section, and submit your "Accept" recommendation.

Reviewer #1: All comments have been addressed

Reviewer #2: All comments have been addressed

Reviewer #3: (No Response)

Reviewer #4: All comments have been addressed

2. Is the manuscript technically sound, and do the data support the conclusions?

Reviewer #1: Yes

Reviewer #2: Partly

Reviewer #3: No

Reviewer #4: Yes

3. Has the statistical analysis been performed appropriately and rigorously? 

Reviewer #1: N/A

Reviewer #2: No

Reviewer #3: No

Reviewer #4: N/A

4. Have the authors made all data underlying the findings in their manuscript fully available?

Reviewer #1: Yes

Reviewer #2: No

Reviewer #3: Yes

Reviewer #4: Yes

5. Is the manuscript presented in an intelligible fashion and written in standard English?

Reviewer #1: Yes

Reviewer #2: Yes

Reviewer #3: Yes

Reviewer #4: Yes

6. Review Comments to the Author

Reviewer #1: I recomend publishing the article in spite of the lack of sufficient statistics, detected by reviewers, based on our experience on extreme environments. I think that the authors have adequately proposed the statistics that can be done in such kind of environments, which is valuable for going ahead.

Reviewer #2: Thanks for the author to the important effort to improve their manuscript which is clearly easier to read.

Based on their feedbacks, I’m still confused about the take-home message and how they provided the sufficiently argued elements.

If I fully understand that the measures on the different sites were done by different teams having their own major goals. However, the variability of the results (image quality , manual/automatic tracking, etc…) could be due to sites or experimentalist. My understanding of providing a biosignature implies a robust and reproducible protocol. For instance L301 : The x, y, t stacks used were usually z projections over multiple planes, but occasionally better SNR was obtained using a single denoised z-plane reconstruction rather than a projection. In Table 2 for the same concentration category, speed is quantified or not among the different sites. In the same idea, claiming an autonomous software package is slightly overselling as authors explain to change of analysis pipeline depending on the SNR of the images.

Active motion is presented as a key element. However, L311 it is mentioned Brownian motion analysis but we don’t understand 1) where are used / shown the results, 2) how they split between directional vs Brownian movement (as for instance NanoTrackJ is done exclusively for diffusive objects). Another question is still how authors split between eukaryotes and prokaryotes only based on their size. Some eukaryotes could be as small as 5µm (Ostreoccucus & others), so for a reader, it could induce erroneous information.

Again, I really appreciate the effort of the authors to improve the text, but several weaknesses in the data analysis seem to me too important to publish under these conditions.

Reviewer #3: I thank the authors for their revisions to the text in response to the reviewers' comments. Unfortunately I cannot recommend the manuscript for publication because my comments about analysis and statistical significance (raised also by reviewer 2) have not been addressed. As I understand, PLOS ONE aims for a high degree of technical rigor, and in its current state the manuscript does not rigorously support its claims.

In particular I remain concerned about the analysis approach, the very small number of samples, and the lack of ground truthing for several sites. The authors note in the revised manuscript that 'Cells were identified as either motile or non-motile via manual observation of the x, y, t stacks. The concentration of observed motile microbes was very low at some of the sites, with a total of only a few organisms identified over many recordings.' They further note that 'The parameters/thresholds used to process the images to increase their trackability are unique for each recording and are chosen through an iterative process of comparing identified localization of particles via the automated tracking algorithms to manual observations of the recordings at each image processing step.'

This approach has several major problems: (1) Relying on manual observation to identify motilility makes the results subject to the particular observer and can introduce bias -- especially considering that the image processing parameters/thresholds were determined on the basis of manual observations. False positive identifications can easily result from such a procedure. (2) The small number of observed motile microbes and the absence of ground truth for some of the data sets means that there is no way to assess what the false positive rate is, and thus no way to verify the claims. (3) Because the criteria for processing the images differ from recording to recording and depend on the intervention of an observer, it would be very difficult to replicate the results in future, independent studies.

I do appreciate the amount of field work and analysis that went into this study, and the aims of the study are well-motivated. But these features alone are not sufficient for publication in PLOS ONE. The technical rigor must be improved.

A minor comment: the figures are overcompressed, and figure 3 in particular is unreadable in my copy of the manuscript.

Reviewer #4: (No Response)

7. PLOS authors have the option to publish the peer review history of their article (what does this mean? ). If published, this will include your full peer review and any attached files.

**Do you want your identity to be public for this peer review?** For information about this choice, including consent withdrawal, please see our Privacy Policy .

Reviewer #1: **Yes: ** Cecilia Dermergasso

Reviewer #2: No

Reviewer #3: No

Reviewer #4: No

---

## [Author Response · Author response to Decision Letter 2]

17 Oct 2024

PLOS ONE Manuscript PONE-D-23-22462

Response to Reviewers

Dear PLOS ONE editor and reviewers,

Thank you for giving us the opportunity to submit a revised draft of the manuscript “Extant Life Detection Using Label-Free Video Microscopy in Analog Aquatic Environments” for publication in the PLOS ONE Journal. We recognize and greatly appreciate the time and labor that you all dedicated to providing input on improving our manuscript and are grateful for the insightful critiques on and valuable improvements to our paper. We have included many of the suggestions made. All changes within the manuscript are seen in red text. Please see below, in blue, for a point-by-point response to the reviewers’ comments and concerns. All page numbers refer to the revised manuscript file with tracked changes.

Reviewers' Comments to the Authors and Respective Responses to Reviewers:

Reviewer 1

I recommend publishing the article in spite of the lack of sufficient statistics, detected by reviewers, based on our experience on extreme environments. I think that the authors have adequately proposed the statistics that can be done in such kind of environments, which is valuable for going ahead.

Author response: Thank you for your comments and approval for publication.

Reviewer 2

Thanks for the author to the important effort to improve their manuscript which is clearly easier to read.

Based on their feedback, I’m still confused about the take-home message and how they provided the sufficiently argued elements.

Again, I really appreciate the effort of the authors to improve the text, but several weaknesses in the data analysis seem to me too important to publish under these conditions.

Author response: Thank you for your kind words as well as taking time to review and suggest further edits. These comments help us strengthen our paper and make it easier for future readers to understand our work.

During this round of edits, we further tried to improve our paper by implementing the following:

Solidifying that the main purpose of the paper is to demonstrate the presence and ability of the DHM to detect active motion, morphology, and optical property biosignatures for a variety of environments.

Adding the argument that limited data sets and ground truthing are similar to what we would expect from off-world experiments. Therefore, knowing what claims can be made with the limited data we have in this paper is useful for future off-world astrobiology missions.

Adding ground truthing for all sites, except one.

Removing characterization of swimming speeds, which adds to the consistency of the quantitative results.

Including displacement comparison of Brownian motion to suspected motility tracks.

Including accuracy metrics as validation for automated HELM method.

Adding the argument for manual tracking and classifying motion as either biotic or abiotic as gold standard for accuracy.

The main take-home message is that active motion, morphology, and optical property biosignatures are detectable from a variety of sites on Earth (mostly comprising of extreme environments that can act as analogs for ocean worlds) using DHM as the imaging technique. We have added text to the introduction, material and methods, and discussion section to make clear that the statement above is the take-home message. In the material methods section, we have added subsections outlining how active motion, morphology, and optical properties are biosignatures and how they were identified. Below is the text that was added to the paper regarding this point.

INTRODUCTION:

L57 –

Techniques for identifying these features of life (i.e. biosignatures) are central to astrobiological goals of detecting and possibly studying life on other planetary bodies. This study investigates the ability of digital holographic microscopy to conduct in situ experiments to detect and classify the prevalence of active motion, morphology, and optical properties of microbial cells as biosignatures of extant life across a variety of extreme environments here on Earth.

L70 –

Active motion in this paper, as detailed in the methods section is characterized as any cell-like object that is moving in a way that is distinct from any known abiotic motion (i.e. Brownian, fluid flow, buoyancy) [5, 7].

With non-motile cells, simple visualization of morphologically complex organisms in natural samples can often be sufficient to conclude that they are alive [5, 8-10]. With smaller organisms —bacteria and archaea—the situation is more complicated. Their cells contain few features to suggest metabolism or complexity, but their optical properties, refractive index and absorbance, alter their contrast uniquely relative to most microminerals present in in situ samples [11]. Therefore, non-motile organisms can be classified by their distinct appearance due to their morphological features or optical properties relative to abiotic objects.

L99 –

The purpose of collecting in situ data from a variety of extreme environments and stimulation experiments here on Earth is to start to build a framework for detecting biosignatures with a DHM instrument for possible future extant life detection missions on a range of planetary body analog environments. All three microbial biosignatures we examined (i.e. active motion, morphology, and optical properties) are present in almost all sites and at least one is present in every site. The hypothesis of this work is that while not every microbe will exhibit detectable motion, morphology, or optical property biosignatures, within most environments – no matter how extreme – some fraction of microbes will be identifiable by these biosignatures. The fraction of motile cells may be increased by stimulation with heat [17], simple nutrients [18], salinity gradients [19], or other chemical or physical attractants or repellents [20. 21].

MATERIAL AND METHODS:

L255 –

4. Biosignatures

Trained experts manually identified and classified active motion, morphology, and optical properties as biosignatures through inspection of x, y, t stacks; x, y images; MHIs; and plotted tracks of particles. False positive rates for manual identification of biosignatures, while theoretically not zero, can be considered negligible [53, 54].

4.1 Active vs. passive motion

Active motion is characterized by distinguishing the motion from known forms of passive motion (e.g. Brownian, fluid flow, buoyancy) [7. 61]. Fluid flow is readily characterized as unidirectional motion of all particles in the field of view, and may be subtracted readily by computer either separately or in conjunction with track analysis. HELM, as well as other particle trackers such as MOSAIC or NTA NanoTrackJ [64], distinguish Brownian motion by plotting root mean squared displacement d vs. time for each tracked particle. Brownian motion scales with the square root of time t:

d= √2Dt (1)

where D is the diffusion coefficient given as,

D=kT⁄6πηr (2).

In (2) k is the Boltzmann constant, T is temperature, η is the viscosity of the fluid, and r is the radius of the particle. Manual distinguishing of Brownian vs. active motion is based upon observation of displacement and is nearly always unambiguous as shown in Fig. 4. At room temperature (25ºC) with a 1 µm radius particle in water, the diffusion coefficient is ~0.24 µm2/s. At 4ºC the viscosity of water is substantially greater, giving a diffusion coefficient of ~0.13 µm2/s. Measured bacterial diffusion coefficients can be 1000-2000 times greater than predicted when cells aggregate [65]. Making it important to ensure that aggregates be distinguished from single cells by imaging, tracking, or both. Rapidly swimming cells (v = 100 µm/s) displace much more rapidly than Brownian particles (Fig. 4A). Even moderate swimming speeds of 10 µm/s are readily distinguished from Brownian motion of single cells; aggregates can be identified by displacement vs. time curves or simply eliminated from manual tracks (Fig. 4B). More slowly swimming cells are distinguished by their displacement vs. time curves (Fig. 4C). We have previously studied Brownian motion of microparticles and non-motile cells in depth and found that displacements match the models well and that displacement vs. time curves can be used to identify passive diffusion [7].

Along with directional drift and displacement-vs.-time curves, HELM also uses other track parameters such as sinuosity and angle of displacement to identify motile particles. In lab monoculture samples, the true positive rate over the false positive rate for automated motile track detection was reported to be 0.92 [61]. Manual results remain superior to any automated results, however, largely due to poor stitching of automatic tracks. Manual tracking combined with plotting displacement curves in the case of slow-moving cells essentially eliminates false positive identification of motile tracks.

The number of observed motile organisms in our samples was highly variable; some sites had only a few organisms identified over many recordings while some had dozens per field of view. This variability led us to use a qualitative analytical approach where motile organisms were classified as either highly abundant (multiple motile cells in each frame), abundant (at least one motile cell per frame), present (motile cells in some but not all frames), or not present (no motile cells in any frame).

Figure 4. Brownian motion vs swimming. (A) Fast swimming, 100 µm/s, is clearly differentiated from Brownian motion even at short time scales of recording. (B) Moderate swimming speeds of 10 µm/s are still many times faster than Brownian motion of single cells and can be differentiated from Brownian motion of aggregates by displacement vs. time curves. (C) The slower swimming and diffusion rates seen at low temperatures can still be readily distinguished, although longer recordings are helpful to capture the shape of the curves.

4.2 Morphology

Morphological features can demonstrate a complexity that is an unambiguous biosignature. Morphology features include organelles, filamentous, appendages, and cellular clustering [5, 9, 10]. The resolution limit of our DHM instrument allows for the identification of organelles in larger eukaryotes. Cells identified to have organelles were classified as “large” cells and were generally >20 µm in diameter. “Small” cells were those <5 µm in diameter where no organelles could be distinguished, and cells could not be classified as eukaryotic or prokaryotic by DHM alone, as the environments studied often contained an abundance of small eukaryotes [66,67].

4.3 Optical properties

Absorbance and refractive index can both be used as biosignatures as both amplitude and phase images allow one to distinguish cells from microminerals. The difference in absorbance and refractive index of minerals, relative to cells, is generally represented in our microscopy images as higher contrast and sharper edges. Trained scientists can use these features alongside morphology and motion to label a particle as likely a cell or not.

Contrast in amplitude images is given by a combination of absorbance and scattering:

σ=σ_abs+σ_scatt (3)

For absorbing particles, the absorbance term is approximately equal to the absorbance A as calculated from the Beer-Lambert Law,

A=ε(λ)ct (4)

where ε(λ) is the extinction coefficient (or imaginary part of the refractive index, a measure of how much light is being absorbed at a given wavelength) of the absorbing molecule, c is its concentration, and t is the thickness of the cell. With our illumination wavelength at 405 nm, chlorophyll is strongly absorbing, so that photosynthetic cells appeared dark. Many minerals, in contrast, are reflective rather than absorbing and appeared bright in amplitude. Small, nearly transparent cells such as bacteria are dominated by the scattering term, which can be approximated using Mie scattering theory or calculated more accurately for non-spherical cells using the discrete dipole approximation.

In phase reconstructions from DHM images, contrast Δϕ is given by

Δϕ=2πtΔn/λ (5)

where λ is the wavelength of illumination, t is the thickness of the specimen, and Δn is the difference in (imaginary) refractive index between the object and its surrounding medium (in this case water, with n = 1.33). Most cells have Δn~0.05, whereas minerals have values 5-10 fold higher [11]. The limit of detection with a single exposure in our system was approximately Δϕ/2π=0.05, or t/λ~1 for detection of cells.

L656 –

Discussion

As indicated in a 2016 NASA report, microscope-based identification of cell-like objects on future missions to other planetary bodies, whether or not the objects are definitively alive, will assist other means of characterization such as mass spectrometry, Raman or infrared spectroscopy, or other chemical analyses [3]. Chemical signals coming from a known self-contained object, or a selection of such objects, will be more compelling as biosignatures when images showing the objects accompany the chemical data contain biosignatures of their own. This study gives context on how to best use a DHM instrument to detect biosignatures in extreme environments. The in situ identification of at least one biosignature (i.e. motion, morphology, or optical properties) was confirmed at every site. Even though not all biosignatures were present in each recording, these results help build a case and framework for how to search for these biosignatures during future extant life detection missions.

Label-free amplitude and phase video microscopy can provide an array of biosignatures, some definitive and others suggestive. Either active motility or clear morphological features, if observed on another planet, would be a compelling biosignature, especially in combination with chemical studies demonstrating the presence of complex molecules and/or disequilibrium. Absorbance and refractive index optical properties are not definitive biosignatures on their own but fall into this suggestive category. The prevalence of motility, morphology, and optical property biosignatures in analog sites of astrobiological significance has remained largely unexplored. In this study, we observed motility and optical property biosignatures in samples from all the environments tested. Morphological features were identified in all environments except for cryopeg, alkaline spring pool, and subglacial void. The slowest microbes moved well above distinguishable speeds from Brownian and other forms of abiotic motion. Simple stimuli resulted in dramatic increases in motility in seawater and sea ice brines. Raising the temperature of sea ice brines from ambient (< 0ºC) to 4ºC resulted in small microbial motility, as did addition of 1 mM L-serine (but not other amino acids or glucose). Gradients of L-serine were not required to observe motility, though they could be used for further confirmation of directed swimming. Larger, clearly eukaryotic organisms were spontaneously motile in all sea ice and brine samples.

Further descriptions of how we implemented the other improvements are detailed in the responses below.

1. If I fully understand that the measures on the different sites were done by different teams having their own major goals. However, the variability of the results (image quality , manual/automatic tracking, etc…) could be due to sites or experimentalist. My understanding of providing a biosignature implies a robust and reproducible protocol. For instance L301 : The x, y, t stacks used were usually z projections over multiple planes, but occasionally better SNR was obtained using a single denoised z-plane reconstruction rather than a projection. In Table 2 for the same concentration category, speed is quantified or not among the different sites. In the same idea, claiming an autonomous software package is slightly overselling as authors explain to change of analysis pipeline depending on the SNR of the images.

Author response: Yes, while it is accurate to state that data collection was done by different teams with their own major goals, it is important to note that image processing and biosignature detection was done in similar manner for each site. By similar we mean that we first attempted to clean up da

---

## [Decision Letter · Decision Letter 2]

26 Dec 2024

PONE-D-23-22462R2Extant life detection using label-free video microscopy in analog aquatic environmentsPLOS ONE

Dear Dr. Snyder,

Thank you for submitting your manuscript to PLOS ONE. After careful consideration, we feel that it has merit but does not fully meet PLOS ONE’s publication criteria as it currently stands. Therefore, we invite you to submit a revised version of the manuscript that addresses the remaining points raised during the review process.

We look forward to receiving your revised manuscript.

Kind regards,

Michael Schubert

Academic Editor

PLOS ONE

Journal Requirements:

Reviewers' comments:

Reviewer's Responses to Questions

**Comments to the Author**

1. If the authors have adequately addressed your comments raised in a previous round of review and you feel that this manuscript is now acceptable for publication, you may indicate that here to bypass the “Comments to the Author” section, enter your conflict of interest statement in the “Confidential to Editor” section, and submit your "Accept" recommendation.

Reviewer #3: (No Response)

Reviewer #4: All comments have been addressed

2. Is the manuscript technically sound, and do the data support the conclusions?

Reviewer #3: Partly

Reviewer #4: Yes

3. Has the statistical analysis been performed appropriately and rigorously? 

Reviewer #3: I Don't Know

Reviewer #4: Yes

4. Have the authors made all data underlying the findings in their manuscript fully available?

Reviewer #3: Yes

Reviewer #4: Yes

5. Is the manuscript presented in an intelligible fashion and written in standard English?

Reviewer #3: Yes

Reviewer #4: Yes

6. Review Comments to the Author

Reviewer #3: I apologize to the authors for the delay in getting them this review. The authors have made many changes in response to my and the other reviewers' comments. In response to my comments, they have now noted why they rely on manual identification. I think the revised justification is fine for the manuscript, which is trying to establish some baseline for future analyses. However, the authors note in Line 256 that "Trained experts manually identified and classified active motion, morphology, and optical properties as biosignatures through inspection of...". In the interests of reproducibility, it will be important for the authors to specify who the experts are and, more importantly, how they were trained. After the authors add this information, I would favor acceptance. I do not need to see the manuscript again.

Reviewer #4: Based on the earlier reviews and revisions, it is clear that there are some shortcomings in the statistics of the observed motion-based biosignatures (small number of observed motile microbes, absence of ground truthing, identifying motility manually instead of automated tracking for verification).

However, I think the authors have now very well explained 1) the exact purpose of this study, 2) the limits of getting good statistics in the extreme environments that were studied, 3) the need for such studies for future space missions, in which similar lack of statistics will likely be a reality.

In the Introduction of the paper (line 105-107) the authors write: “The hypothesis of this work is that while not every microbe will exhibit detectable motion, morphology, or optical property biosignatures, within most environments – no matter how extreme – some fraction of microbes will be identifiable by these biosignatures”. I think this is the important message, and makes this study relevant for future space missions.

Overall, the authors made every effort to provide ground truthing in the form of fluorescence microscopy and in some cases gene sequencing. They presented many DHM images, and videos. Applying the method to in situ samples in extreme environments, rather than ideal lab culture samples, is in my view a very important test. In that sense it forms a more realistic basis for future space missions.

So overall, although I recognize the shortcomings pointed out by the other reviewers (particularly with respect to statistical rigor), I think this study warrants publication. It is a very valuable basis for future studies that can add similar data and thereby increasing the statistics of this method.

7. PLOS authors have the option to publish the peer review history of their article (what does this mean? ). If published, this will include your full peer review and any attached files.

**Do you want your identity to be public for this peer review?** For information about this choice, including consent withdrawal, please see our Privacy Policy .

Reviewer #3: No

Reviewer #4: No

---

## [Author Response · Author response to Decision Letter 3]

11 Jan 2025

PLOS ONE Manuscript PONE-D-23-22462

Response to Reviewers

Dear PLOS ONE editor and reviewers,

Thank you for giving us the opportunity to submit a revised draft of the manuscript “Extant Life Detection Using Label-Free Video Microscopy in Analog Aquatic Environments” for publication in the PLOS ONE Journal. We recognize and greatly appreciate the time and labor that you all dedicated to providing input on improving our manuscript and are grateful for the insightful critiques on and valuable improvements to our paper. We have included many of the suggestions made. All changes within the manuscript are seen in red text. All page numbers refer to the revised manuscript file with tracked changes.

Reviewers' Comments to the Authors and Respective Responses to Reviewers:

Reviewer #3:

I apologize to the authors for the delay in getting them this review. The authors have made many changes in response to my and the other reviewers' comments. In response to my comments, they have now noted why they rely on manual identification. I think the revised justification is fine for the manuscript, which is trying to establish some baseline for future analyses. However, the authors note in Line 256 that "Trained experts manually identified and classified active motion, morphology, and optical properties as biosignatures through inspection of...". In the interests of reproducibility, it will be important for the authors to specify who the experts are and, more importantly, how they were trained. After the authors add this information, I would favor acceptance. I do not need to see the manuscript again.

Author response: Thank you for your thoughtful feedback and for taking the time to review our manuscript. We greatly appreciate your constructive suggestion to enhance the reproducibility of our work. We have outlined in the “Supporting Information” document specifically how researchers were trained to identify the biosignatures outlined in the paper and who did this analysis for our publication. This revised text is found below.

L205 –

Biosignature Identification Training

The experts involved in identifying and classifying active motion, morphology, and optical properties as biosignatures were individuals with advanced academic and professional backgrounds in relevant fields, e.g., microbiology, microscopy, biophysics, etc. The training process consisted of sessions that covered identification of active motion, morphology, and optical properties as biosignatures. These sessions included both theoretical instruction and practical exercises using multiple recordings of lab grown monocultures of different sized organisms and a subset of experimental data to ensure a consistent understanding of the classification criteria. To ensure high reliability, the experts underwent an initial calibration phase, during which they independently classified a set of samples, and their results were compared and discussed in group meetings. Discrepancies in classification were resolved through collaborative review and consensus-building, ensuring that the experts were aligned in their interpretation of the biosignatures. The final classification process was carried out under the established protocols outlined in the main text of this paper to ensure consistency across different experts. Biosignature identification and analysis of sites were performed by researchers Carl Snyder, Dr. Manuel Bedrossian, Dr. Casey Barr, and Dr. Jay Nadeau.

Reviewer #4:

Based on the earlier reviews and revisions, it is clear that there are some shortcomings in the statistics of the observed motion-based biosignatures (small number of observed motile microbes, absence of ground truthing, identifying motility manually instead of automated tracking for verification).

However, I think the authors have now very well explained 1) the exact purpose of this study, 2) the limits of getting good statistics in the extreme environments that were studied, 3) the need for such studies for future space missions, in which similar lack of statistics will likely be a reality.

In the Introduction of the paper (line 105-107) the authors write: “The hypothesis of this work is that while not every microbe will exhibit detectable motion, morphology, or optical property biosignatures, within most environments – no matter how extreme – some fraction of microbes will be identifiable by these biosignatures”. I think this is the important message, and makes this study relevant for future space missions.

Overall, the authors made every effort to provide ground truthing in the form of fluorescence microscopy and in some cases gene sequencing. They presented many DHM images, and videos. Applying the method to in situ samples in extreme environments, rather than ideal lab culture samples, is in my view a very important test. In that sense it forms a more realistic basis for future space missions.

So overall, although I recognize the shortcomings pointed out by the other reviewers (particularly with respect to statistical rigor), I think this study warrants publication. It is a very valuable basis for future studies that can add similar data and thereby increasing the statistics of this method.

Author response: Thank you for your kind words as well as taking time to review and suggest further edits. These comments help us strengthen our paper and made it easier to understand our work.

---

## [Editor Report · Decision Letter 3]

14 Jan 2025

Extant life detection using label-free video microscopy in analog aquatic environments

PONE-D-23-22462R3

Dear Dr. Snyder,

We’re pleased to inform you that your manuscript has been judged scientifically suitable for publication and will be formally accepted for publication once it meets all outstanding technical requirements.

Kind regards,

Michael Schubert

Academic Editor

PLOS ONE

---

## [Editor Report · Acceptance letter]

PONE-D-23-22462R3

PLOS ONE

Dear Dr. Snyder,

I'm pleased to inform you that your manuscript has been deemed suitable for publication in PLOS ONE. Congratulations! Your manuscript is now being handed over to our production team.

Kind regards,

on behalf of

Dr. Michael Schubert

Academic Editor

PLOS ONE